# Regional soil erosion assessment based on sample survey and geostatistics

Shuiqing Yin[1], Zhengyuan Zhu[2], Li Wang[2], Baoyuan Liu[1], Yun Xie[1], Guannan Wang[3] and Yishan Li[1]

[1]State Key Laboratory of Earth Surface Processes and Resource Ecology, Faculty of Geographical Science, Beijing Normal University, Beijing 100875, China

[2]Department of Statistics, Iowa State University, Ames 50010, USA

[3]Department of Mathematics, College of William & Mary, Williamsburg 23185, USA

*Correspondence to*: Baoyuan Liu (baoyuan@bnu.edu.cn)

**Abstract.** Soil erosion is one of the most significant environmental problems in China. From 2010-2012, the fourth national census for soil erosion sampled 32,364 Primary Sampling Units (PSUs, small watersheds) with the areas of 0.2-3 $km^2$. Land use and soil erosion controlling factors including rainfall erosivity, soil erodibility, slope length, slope steepness, biological practice, engineering practice, and tillage practice for the PSUs were surveyed, and soil loss rate for each land use in the PSUs were estimated using an empirical model Chinese Soil Loss Equation (CSLE). Though the information collected from the sample units can be aggregated to estimate soil erosion conditions on a large scale, the problem of estimating soil erosion condition on a regional scale has not been well addressed. The aim of this study is to introduce a new model-based regional soil erosion assessment method combining sample survey and geostatistics. We compared seven spatial interpolation models based on Bivariate Penalized Spline over Triangulation (BPST) method to generate a regional soil erosion assessment from the PSUs. Shaanxi province (3,116 PSUs) in China was used to conduct the comparison and assessment as it is one of the areas with the most serious erosion problem. Ten-fold cross validation based on the PSU data showed the model assisted by the land use, rainfall erosivity factor (R), soil erodibility factor (K), slope steepness factor (S) and slope length factor (L) derived from 1:10000 topography map is the best one, with the model efficiency coefficient (ME) being 0.75 and the MSE being 55.8% of that for the model assisted by the land use alone. Among four erosion factors as the covariates, S factor contributed the most information, followed by K and L factors, and R factor made almost no contribution to the spatial estimation of soil loss. LS-factor derived from 30-m or 90-m SRTM DEM data worsened the estimation when they were used as the

covariates for the interpolation of soil loss. Due to the unavailability of 1:10000 topography map for the entire area in this study, the model assisted by the land use, R and K factor with a resolution of 250 m was used to generate the regional assessment of the soil erosion for Shaanxi province. It demonstrated that 54.3% of total land in Shaanxi province had annual soil loss equal to or greater than 5 t ha$^{-1}$ y$^{-1}$. High (20-40 t ha$^{-1}$ y$^{-1}$), severe (40-80 t ha$^{-1}$ y$^{-1}$) and extreme (>80 t ha$^{-1}$ y$^{-1}$) erosion occupied 14.0% of the total land. The dry land and irrigated land, forest, shrub land and grassland in Shaanxi province had mean soil loss rates of 21.77, 3.51, 10.00, and 7.27 t ha$^{-1}$ y$^{-1}$, respectively. Annual soil loss was about 207.3 Mt in Shaanxi province, with 68.9% of soil loss originating from the farmlands and grasslands in Yan'an and Yulin districts in the northern Loess Plateau region and Ankang and Hanzhong districts in the southern Qingba mountainous region. This methodology provides a more accurate regional soil erosion assessment and can help policy-makers to take effective measures to mediate soil erosion risks.

## 1 Introduction

With a growing population and a more vulnerable climate system, land degradation is becoming one of the biggest threats to food security and sustainable agriculture in the world. Two of the primary sources of land degradation are water and wind erosion (Blanco and Lal, 2010). To improve the management of soil erosion and aid policy-makers to take suitable remediation measures and mitigation strategies, the first step is to monitor and assess the related system to obtain timely and reliable information about soil erosion conditions under present climate and land use. Assessments on the risks of soil erosion under different scenarios of climate change and land use are also very important (Kirkby et al., 2008).

Scale is a critical issue in soil erosion modeling and management (Renschler and Harbor, 2002). When the spatial scale is small, experimental runoff plots, soil erosion markers (e.g. Caesium 137) or river sediment concentration measurement devices (e.g. optical turbidity sensors) are useful tools. However, when the regional scale is considered, it is impractical to measure soil loss across the entire region. A number of approaches have been used to assess the regional soil erosion in different countries and regions over the world, such as expert-based factorial scoring, plot-based, field-based and model-based assessments, and so on.

Factorial scoring was used to assess soil erosion risk when erosion rates were not required, and one only needs a spatial distribution of erosion (Guo and Li, 2009; Le Bissonnais et al., 2001). The classification or scoring of erosion factors (e.g. land use, rainfall erosivity, soil erodibility and slope) into discrete classes and the criteria

used to combine the classes are based on expert experience. The resulting map depicts classes ranging from very
low to very high erosion or erosion risk. However, the factorial scoring approach has limitations on subjectivity
and qualitative characteristics (Morgan, 1995; Grimm et al., 2002). A plot-based approach extrapolated the
measurements from runoff plots to the region (Gerdan et al., 2010; Guo et al., 2015). However, Gerdan et al.
(2010) discussed that the direct extrapolation may lead to poor estimation of regional erosion rates if the scale
issue is not carefully taken into consideration. Evans et al., (2015) recommended a field-based approach,
combining visual interpretations of aerial and terrestrial photos and direct field survey of farmers' fields in
Britain. However, its efficiency, transparency and accuracy were questioned (Panagos et al., 2016a).
The model-based approach can not only assess soil loss up to the present time, but also has the advantage of
assessing future soil erosion risk under different scenarios of climate change, land use and conservation
practices (Kirkby et al., 2008; Panagos et al., 2015b). USLE (Wischmeier and Smith, 1965; Wischmeier and
Smith, 1978) is an empirical model based on the regression analyses of more than 10,000 plot-years of soil loss
data in the USA and is designed to estimate long-term annual erosion rates on agricultural fields. (R)USLE
(Wischmeier and Smith, 1978; Renard et al., 1997; Foster, 2004) and other adapted versions (for example,
Chinese Soil Loss Equation, CSLE, Liu et al., 2002), are the most widely used models in the regional scale soil
erosion assessment due to relative simplicity and robustness (Singh et al., 1992; Van der Knijff et al., 2000; Lu
et al., 2001; Grimm et al., 2003; Liu, 2013; Bosco et al., 2015; Panagos et al., 2015b).
The applications of USLE and its related models in the assessment of regional soil erosion can be generally
grouped into three categories. The first category is the area sample survey approach. One representative is the
National Resource Inventory (NRI) survey on U.S. non-Federal lands (Nusser and Goebel, 1997; Goebel, 1998;
Breidt and Fuller, 1999). USDA-NRCS (2015) summarized the results from the 2012 NRI, which also included
a description of the NRI methodology and use. A summary of NRI results on rangeland is presented in Herrick
et al. (2010). See for example Brejda et al. (2001) and Hernandez, et al. (2013) for some applications using NRI
data. Since a rigorous probability based area sampling approach is used to select the sampling sites, the design
based approach is robust and reliable when it is used to estimate the soil erosion at the national and state level.
However, due to sample size limitations, estimates at the sub-state level are more uncertain.
The second category is based on the multiplication of erosion factor raster layers. Each factor in the (R)USLE
model is a raster layer and soil loss was obtained by the multiplication of numerous factors, which was usually
conducted under GIS environment (Lu et al. 2001; Bosco et al., 2015; Panagos et al., 2015b; Ganasri and
Ramesh, 2015; Rao et al., 2015; Bahrawi et al., 2016). A European water erosion assessment which introduced
high-resolution (100 m) input layers reported the result that the mean soil loss rate in the European Union's
erosion-prone lands was 2.46 t ha$^{-1}$ y$^{-1}$ (Panagos et al., 2015b). This work is scientifically controversial mainly
due to questions on three aspects: (1) Should the assessment be based on the model simulation or the field
survey? (2) Are the basic principles of the (R)USLE disregarded? and (3) Are the estimated soil loss rates
realistic (Evans and Boardman, 2016; Fiener and Auerswald, 2016; Panagos et al., 2016a, b)? Panagos et al.
(2016a, 2016b) argued that the field survey method proposed by Evans et al. (2015) was not suitable for the
application at the European scale mainly due to work force and time requirements. They emphasized their work
focused on the differences and similarities between regions and countries across the Europe and RUSLE model
with the simple transparent structure was able to meet the requirements if harmonized datasets were inputted.
The third category is based on the sample survey and geostatistics. One example is the fourth census on soil
erosion in China during 2010-2012, which was based on a stratified unequal probability systematic sampling
method (Liu et al., 2013). In total, 32,364 Primary Sampling Units (PSUs) were identified nationwide to collect
factors for water erosion prediction (Liu, 2013). CSLE was used to estimate the soil loss for the PSUs. A spatial
interpolation model was used to estimate the soil loss for the non-sampled sites.
Remote sensing technique has unparalleled advantage and potential in the work of regional scale soil erosion
assessment (Veirling, 2006; Le Roux et al., 2007; Guo and Li, 2009; Mutekanga et al., 2010; El Haj El Tahir et
al., 2010). The aforementioned assessment method based on the multiplication of erosion factors under GIS
interface was largely dependent on the remote sensing dataset (Panagos et al., 2015b; Ganasri and Ramesh,
2015; Bahrawi et al., 2016), which also provided important information for the field survey work. For example,
NRI relied exclusively on the high resolution remote sensing images taken from fixed wing airplanes to collect
land cover information. However, many characteristics of soil erosion cannot be derived from remote sensing
images. Other limitations include the accuracy of remote sensing data, the resolution of remote sensing images,
financial constraints and so on, which result in some important factors influencing soil erosion being not
available for the entire domain. It is important to note that the validation is necessary and required to evaluate
the performance of a specific regional soil erosion assessment method, although the validation process is
difficult to implement in the regional scale assessment and is not well addressed in the existing literature (Gobin
et al., 2004; Vrieling, 2006; Le Roux et al., 2007; Kirkby, et al., 2008).
An important issue in the regional soil erosion assessment based on survey sample is how to infer the soil
erosion conditions including the extent, spatial distribution and intensity for the entire domain from the
information of PSUs. NRI used primarily a design based approach to estimate domain level statistics.
While robust and reliable for large domains which contain enough sample sites, such method has large
uncertainties when it was used for the small domain. The method to obtain domain level statistics used in
the fourth census of soil erosion in China was different from that used by NRI. A simple spatial model was used
to smooth the proportion of soil erosion directly in China, which is an attempt to interpolate sample survey units
information using geostatistics. The land use is one of the critical pieces of information in the soil erosion
assessment (Ganasri and Ramesh, 2015), which is available for the entire domain. The erosion factors rainfall
erosivity (R) and soil erodibility (K) are also available for the entire domain. The slope length (L) and slope
degree (S) factors can be derived from 30-m and 90-m Digital Elevation Model (DEM) data from shuttle radar
topography mapping mission (SRTM). The other factors including the biological (B), engineering (E) and tillage
(T) practice factors are either impossible or very difficult to obtain for the entire region at this stage. We
sampled small watersheds (PSUs) to collect detailed topography information (1:10000 topography map with 5-
m contour intervals) and conducted field survey to collect soil and water conservation practice information. The
purpose of this study is to introduce a new regional soil erosion assessment method which combines data from
the sample survey with factor information over the entire domain using geostatistics. We compare seven semi-
parametric spatial interpolation models assisted by land use and single or multiple erosion factors based on
bivariate penalized spline over triangulation (BPST) method to generate regional soil loss (A) assessment from
the PSUs. A sensitivity analysis of topography factor derived from different resolutions of DEM data was also
conducted. There are 3116 PSUs in the Shaanxi province and its surrounding areas which were used as an
example to conduct the comparison and demonstrate assessment procedures (Fig. 1). For many regions in the
world, data used to derive erosion factor such as conservation practice factor is often not available for all area,
or the resolution is not adequate for the assessment. Therefore, the assessment method combining sample survey
and geostatistics proposed in this study is valuable.
**2 Data and Methods**
**2.1 Sample and field survey**
The design of the fourth census on soil erosion in China is based on a map with Gauss–Krüger projection, where
the whole of China was divided into 22 zones with each zone occupying three longitude degrees width (From
central meridian towards west and east 1.5 degrees each). Within each zone, beginning from the central meridian
and the equator, we generated grids with a size of 40 km $\times$ 40 km (Fig. 2), which are the units at the first level
(County level). The second level is Township level with a size of 10 km $\times$ 10 km. The third level is the control
area, with a size of 5 km $\times$ 5 km. The fourth level is the 1 km $\times$ 1 km grid located in the middle of the control
area. The 1 km $\times$ 1 km grid is the PSU in the plain area, whereas in the mountainous area, a small watershed
with area between 0.2-3 km$^2$ which also intersects with the fourth level 1 km $\times$ 1 km grid was randomly picked
as the PSU. The area for the mountainous PSU is restricted to be between 0.2-3 km$^2$, which is large enough for
the enumerator and not too large to be feasible to conduct field work. There is a PSU within every 25 km$^2$,
which suggests the designed sample density is about 4%. In practice, due to the limitation of financial resources,
the surveyed sample density is 1% for most mountainous areas. The density for the plain area is reduced to
0.25% due to the lower soil erosion risk (Li et al., 2012).
The field survey work for each PSU mainly included: (1) recording the latitude and longitude information for
the PSU using a GPS; (2) drawing boundaries of plots in a base map of the PSU; (3) collecting the information
of land use and soil conservation measures for each plot; and (4) taking photos of the overview of PSUs, plots
and soil and water conservation measures for future validation. A plot was defined as the continuous area with
the same land use, the same soil and water conservation measures, and the same canopy density and vegetation
fraction in the PSU (difference <=10%, Fig. 3). For each plot, land use type, land use area, biological measures,
engineering measures and tillage measures were surveyed. In addition, vegetation fraction was surveyed if the
land use is a forest, shrub land or grassland. Canopy density is also surveyed if the land use is a forest.
**2.2 Database of PSUs in Shaanxi and its surrounding areas**
A convex hull of the boundary of Shaanxi province was generated, with a buffer area of 30 km outside of
the convex hull (Fig. 4). The raster of R factor, K factor and 1:100000 land use map with a resolution of
250$\times$250 m pixels for the entire area were collected. PSUs located inside the entire area were used, which
included 1775 PSUs in the Shaanxi province and 1341 PSUs from the provinces surrounding the Shaanxi
province, including Gansu (430), Henan (112), Shanxi (345), Inner Mongolia (41), Hubei (151),
Chongqing (55), Sichuan (156) and Ningxia (51). There were 3116 PSUs in total. We had the information
of longitude and latitude, land use type, land use area and factor values of R, K, L, S, B, E and T for each
plot of the PSU. The classification system of the land use for the entire area and that for the survey units
were not synonymous with each other. Rather, they were grouped into eleven land use types include (1)
paddy, (2) dry land & irrigated land, (3) orchard & garden, (4) forest, (5) shrub land (6) grassland, (7)
water body, (8) construction land, (9) transportation land, (10) bare land and (11) unused land such as
sandy land, Gebi and uncovered rock to make them correspond to each other.
**2.3 Soil loss estimation for the plot, land use and PSU**
Soil loss for a plot can be estimated using CSLE equation as follows:
$$A_{uk} = R_{uk} \cdot K_{uk} \cdot L_{uk} \cdot S_{uk} \cdot B_{uk} \cdot E_{uk} \cdot T_{uk},$$    (1)
where $A_{uk}$ is the soil loss for the $k^{th}$ plot with the land use u (t ha$^{-1}$ y$^{-1}$), $R_{uk}$ is the rainfall erosivity (MJ mm
ha$^{-1}$ h$^{-1}$ y$^{-1}$), $K_{uk}$ is the soil erodibilty (t ha h MJ$^{-1}$ ha$^{-1}$ mm$^{-1}$), $L_{uk}$ is the slope length factor, $S_{uk}$ is the
slope steepness factor, $B_{uk}$ is the biological practice factor, $E_{uk}$ is the engineering practice factor, $T_{uk}$ is
the tillage practice factor. The definitions of A, R and K are similar to that of USLE. Biological (B),
Engineering (E) and Tillage (T) factor is defined as the ratio of soil loss from the actual plot with
biological, engineering or tillage practices to the unit plot. Biological practices are the measures to increase
the vegetation coverage for reducing runoff and soil loss such as trees, shrubs and grass plantation and
natural rehabilitation of vegetation. Engineering practices refer to the changes of topography by
engineering construction on both arable and non-arable land using non-normal farming equipment (such as
earth mover) for reducing runoff and soil loss such as terrace, check dam and so on. Tillage practices are
the measures taken on the arable land during ploughing, harrowing and cultivation processes using normal
farming operations for reducing runoff and soil loss such as crop rotation, strip cropping and so on (Liu et
al., 2002).
Liu et al. (2013) introduced the data and methods for calculating each factor. Here we present a brief
introduction. Land use map with a scale of 1:100000 is from China's Land Use/cover Datasets (CLUD), which
were updated regularly at a five-year interval from the late 1980s through the year of 2010 with standard
procedures based on Landsat TM/ETM images (Liu et al., 2014). The land use map used in this study was the
2010 version (Fig. 5a). 2678 weather and hydrologic stations with erosive daily rainfall from 1981 through 2010
were collected and used to generate the R factor raster map over the entire China (Xie et al., 2016). And for the
K factor, soil maps with scales of 1:500,000 to 1:200,000 (for different provinces) from the Second National
Soil Survey in 1980s generated more than 0.18 million polygons of soil attributes over mainland China, which
was the best available spatial resolution of soil information we could collect at present. The physicochemical
data of 16,493 soil samples (belong to 7764 soil series, 3366 soil families, 1597 soil subgroups and 670 soil
groups according to Chinese Soil Taxonomy) from the maps and the latest soil physicochemical data of 1065
samples through the ways of field sampling, data sharing and consulting literatures were collected to generate
the K factor for the entire country (Liang et al., 2013; Liu et al., 2013). We assumed the result of the soil survey
could be used to estimate the K factor in our soil erosion survey. R factor raster map for the study area was
clipped from the map of the country as well as the K factor raster map (Fig. 5b, c). Topography contour map
with a scale of 1:10000 for PSUs were collected to derive the slope lengths and slope degrees and to calculate
the slope length factors and slope steepness factors (Fu et al., 2013). Topography contour maps with a scale of
1:10000 for the entire region were not available at present. Fig. 5d was based on SRTM 90-m DEM dataset and
it was used to demonstrate the variation in the topography. The land use map was used to determine the
boundary of forest, shrub, and grass land. For these three land use types, MODIS NDVI and HJ-1 NDVI were
combined to derive vegetation coverage. For the shrub and grass land, an assignment table was used to assign a
value of the half-month B factor based on their vegetation coverage; For the forest land, the vegetation coverage
derived from the aforementioned remote sensing data was used as the canopy density, which was combined with
the vegetation fraction under the trees collected during the field survey to estimate the half-month B factor. The
B factor for the whole year was weight-averaged by a weight of rainfall erosivity ratio for this half-month. Both
C factor in Panagos et al. (2015a) and B factor in this study for forest, shrub land and grassland were estimated
based on the vegetation density derived from satellite images. The difference is that C factor in Panagos et al.
(2015a) for arable land and non-arable land was estimated separately based on different methodologies, whereas
in this study, B factor was used to reflect biological practices on the forest, shrub land or grassland for reducing
runoff and soil loss and T factor was used to reflect tillage practices on the farmland for reducing runoff and soil
loss. For the farmland, biological factor equals 1 and for the other land uses, tillage factor equals 1. The
engineering practice factor and tillage practice factor were assigned values based on the field survey and
assignment tables for different engineering and tillage measures, which were obtained from published references
(Guo et al., 2015).
In a PSU, there may be several plots within the same land use. Soil loss for the same land use was weight-
averaged by the area of the plots with the same land use:
$$A_{ui} = \frac{\sum_{k=1}^{q}(A_{uik}S_{uik})}{\sum_{k=1}^{q}S_{uik}}, \qquad (2)$$
where $A_{ui}$ is the average soil loss for the land use u in the sample unit i (t ha$^{-1}$ y$^{-1}$); $A_{uik}$ is the soil loss for
the plot k with the land use u (t ha$^{-1}$ y$^{-1}$); $S_{uik}$ is the area for the plot k with the land use u (ha). $q$ is the
number of plots with the land use u in the unit i.
**2.4 Seven spatial models based on BPST method**
**2.4.1 Seven spatial models**
Model I: Estimating A with the land use as the auxiliary information. For the water body, transportation
land and unused area, the estimation of soil loss for the u$^{th}$ land use and j$^{th}$ pixel $\hat{A}_{uj}$ was set to be zero. For
the rest of the land use types, $A_{ui}$ for each land use was interpolated separately first and soil loss values
for the entire domain $\hat{A}_{uj}$ are the combination of estimation for all land uses.
Model II: Estimating A with R and land use as the auxiliary information. For each sampling unit *i* in land
use *u*, define
$$Q_{ui} = \frac{A_{ui}}{R_{ui}}, \tag{3}$$
where $R_{ui}$ is the rainfall erosivity value. For land use *u*, we smooth $Q_{ui}$'s over the entire domain using the
longitude and latitude information, and obtain the estimator $\hat{Q}_{uj}$ of $Q_{uj}$ for every pixel *j*. Then, for the j$^{th}$
pixel in land use *u*, we estimate the soil loss $A_{uj}$ by
$$\hat{A}_{uj} = \hat{Q}_{uj} \cdot R_{uj}, \tag{4}$$
Model III: Estimating A with K and land use as the auxiliary information. This model is similar to Model
II, except that we use $K_{ui}$ instead of $R_{ui}$ in equation (3) and $K_{uj}$ instead of $R_{uj}$ in equation (4).
Model IV: Estimating A with L and land use as the auxiliary information. This model is similar to Model II,
except that we use $L_{ui}$ instead of $R_{ui}$ in equation (3) and $L_{uj}$ instead of $R_{uj}$ in equation (4).
Model V: Estimating A with S and land use as the auxiliary information. This model is similar to Model II,
except that we use $S_{ui}$ instead of $R_{ui}$ in equation (3) and $S_{uj}$ instead of $R_{uj}$ in equation (4).
Model VI: Estimating A with R, K and land use as the auxiliary information. This model is similar to
Model II, except that we use $R_{ui}K_{ui}$ instead of $R_{ui}$ in equation (3) and $R_{uj}K_{uj}$ instead of $R_{uj}$ in
equation (4).
Model VII: Estimating A with R, K, L, S and land use as the auxiliary information. This model is similar to
Model II, except that we use $R_{ui}K_{ui}L_{ui}S_{ui}$ instead of $R_{ui}$ in equation (3) and $R_{uj}K_{uj}L_{uj}S_{uj}$ instead of
$R_{uj}$ in equation (4).
**2.4.2 Bivariate penalized spline over triangulation method**
In spatial data analysis, there are mainly two approaches to make the prediction of a target variable. One approach
(e.g., kriging) treats the value of a target variable at each location as a random variable and uses the covariance
function between these random variables or a variogram to represent the correlation; another approach (e.g., spline
or wavelet smoothing) uses a deterministic smooth surface function to describe the variations and connections
among values at different locations. In this study, Bivariate Penalized Spline over Triangulation (BPST), which
belongs to the second approach, was used to explore the relationship between location information in a two-
dimensional (2-D) domain and the response variable. The BPST method we consider in this work has several
advantages. First, it provides good approximations of smooth functions over complicated domains. Second, the
computational cost for spline evaluation and parameter estimation are manageable. Third, the BPST doesn't
require the data to be evenly distributed or on a regular-spaced grid.
To be more specific, let $(x_i, y_i) \in \Omega$ be the latitude and longitude of unit i for $i = 1, 2, ..., n$. Suppose we observe
$z_i$ at locations $(x_i, y_i)$ and $\{(x_i, y_i, z_i)\}_{i=1}^{n}$ satisfy
$z_i = f(x_i, y_i) + \varepsilon_i, i = 1, 2, ..., n,$        (5)
where $\varepsilon_i's$ are random variables with mean zero, and $f(x_i, y_i)$ is some smooth but unknown bivariate
function. To estimate f, we adopt the bivariate penalized splines over triangulations to handle irregular
domains. In the following we discuss how to construct basis functions using bivariate splines on a
triangulation of the domain $\Omega$. Details of various facts about bivariate splines stated in this section can be
found in Lai and Schumaker (2007). See also Guillas and Lai (2010) and Lai and Wang (2013) for
statistical applications of bivariate splines on triangulations.
A triangulation of $\Omega$ is a collection of triangles $\Delta = \{\tau_1, \tau_2, ..., \tau_N\}$ whose union covers $\Omega$. In addition, if
a pair of triangles in $\Delta$ intersects, then their intersection is either a common vertex or a common edge. For a
given triangulation $\Delta$, we can construct Bernstein basis polynomials of degree p separately on each
triangle, and the collection of all such polynomials form a basis. In the following, let $S_r^p(\Delta)$ be a spline
space of degree p and smoothness r over triangulation $\Delta$. Bivariate B-splines on the triangulation are
piecewise polynomials of degree p (polynomials on each triangle) that are smoothly connected across
common edges, in which the connection of polynomials on two adjacent triangles is considered smooth if
directional derivatives up to the $r^{th}$ degree are continuous across the common edge.
To estimate f, we minimize the following penalized least square problem:
$\min_{f \in S_r^p(\Delta)} \left(z_i - f(x_i, y_i)\right)^2 + \lambda PEN(f),$         (6)
Where $\lambda$ is the roughness penalty parameter, and PEN(f) is the penalty given below:
$PEN(f) = \int_{\tau \in \Delta} \left(\frac{\partial^2 f(x,y)}{\partial x^2}\right)^2 + \left(\frac{\partial^2 f(x,y)}{\partial x \, \partial y}\right)^2 + \left(\frac{\partial^2 f(x,y)}{\partial y^2}\right)^2 dxdy,$     (7)
For Models I-VII defined in Section 2.4.1, we consider the above minimization to fit the model, and obtain
the smoothed surface using the data Q and their corresponding location information.
**2.5 Assessment methods**
Mean squared prediction error (MSE) and Nash-Sutcliffe model efficiency coefficient (ME) are used to assess
the performance of models. We estimate the out-of-sample prediction errors of each method using the ten-fold
cross validation. We randomly split all the observations over the entire domain (with the buffer zone) into ten
roughly equal-sized parts. For each t = 1, 2, …., 10, then we leave out part t, fit the model using the other nine
parts (combined) inside the boundary with the buffer zone, and then obtain predictions for the left-out $t^{th}$ part
inside the boundary of Shaanxi Province. The overall mean squared prediction error ($MSE_{overall}$) is calculated by
the average of the sum of the product of individual $MSE_u$ and the corresponding sample size. We first calculated
the MSE of land each use u, u = 1, 2, $\cdots$, 11:
$MSE_u = \frac{\sum_{t=1}^{10} SSE_t}{10},$         (8)
where $SSE_t$ is the sum of squared prediction errors for the $t^{th}$ part. Then, the overall MSE can be calculated using
$MSE_{overall} = \frac{\sum_{u=1}^{11} MSE_u * C_u}{\sum_{u=1}^{11} C_u}.$         (9)
where $C_u$ is the sample size for the land use *u*.
Model efficiency coefficient $ME_u$ for the land use *u* is calculated as follows (Nash and Sutcliffe, 1970):
$$ME_u = 1 - \frac{\sum_{i}^{C_u} [A_{pre,u}(i) - A_{obs,u}(i)]^2}{\sum_{i}^{C_u} [A_{obs,u}(i) - \overline{A_{obs,u}}(i)]^2}$$     (10)
$A_{pre,u}(i)$ and $A_{obs,u}(i)$ are the predicted and observed soil loss for the plot *i* for land use *u*. $ME_{overall}$ stands for
the overall model efficiency by pooling all samples for different land uses together. The ME compares the
simulated and observed values relative to the line of perfect fit. The maximum possible value of ME is 1,
and the higher the value the better the model fit. An efficiency of $ME < 0$ indicates that the mean of the
observed soil loss is a better predictor of the data than the model. The soil loss rate is divided into six soil
erosion intensity levels, which were mild (less than 5 t $ha^{-1}y^{-1}$), slight (5-10 t $ha^{-1}y^{-1}$), moderate (10-20 t $ha^{-1}y^{-1}$),
high (20-40 t $ha^{-1}y^{-1}$), severe (40-80 t $ha^{-1}y^{-1}$), and extreme (no less than 80 t $ha^{-1}y^{-1}$), respectively. Each pixel in
the entire domain was classified into an intensity level according to $A_{uj}$. The proportion of intensity levels, soil
loss rates for different land uses and the spatial distribution of soil erosion intensity levels were computed based
on the soil erosion conditions of pixels located inside of the Shaanxi boundary.
**2.6 Sensitivity analysis of topography factors derived from different resolutions of DEM on the regional**
**soil loss estimation**
Previous research has suggested topography factors should be derived from high resolution topography
information (such as 1:10000 or topography contour maps with finer resolutions, Thomas et al., 2015).
Topography factors based on topography maps with coarser resolutions (such as 1:50000 or 30-m DEM) in the
mountainous and hilly areas have large uncertainties (Wang et al., 2016). Topography contour maps with a scale
of 1:10000 for the entire region were not available at present. To detect if coarser resolution of topography data
available for the entire region, such as SRTM 30-m DEM and 90-m DEM, can be used as the covariate in the
interpolation process, L and S factor were derived from 30-m DEM and 90-m DEM data, respectively (Fu et al.,
2013). The L and S factors derived from 1:10000 topography map for PSUs were used for the cross validation
analysis of Model IV, V and VII to determine the relative contribution of erosion factors as the covariates to the
spatial estimation of soil loss. The L and S factors generated from 30-m and 90-m DEM data, together with
those generated from 1:10000 topography map, were used for the sensitivity analysis based on Model VII.
$MSE_u$ and $MSE_{all}$ based on Eqs. (8) and (9) were used to assess the effect of DEM resolution, from which
topography factors were derived, on the interpolation accuracy of soil loss.
**3 Results**
**3.1 Comparison of MSEs and MEs for seven models and sensitivity of DEM resolution on the MSEs**
Table 1 summarized the MSEs of the soil loss estimation based on different methods. Model VII assisted by R,
K, L, S and land use generated the least overall MSE values and the best result, when L and S were derived
based on 1:10000 topography map. MSE for Model VII was 55.8% of that for Model I. The comparison of four
models with single erosion factor as the covariate (Model II, III, IV and V) showed S factor is the best covariate,
with $MSE_{overall}$ for Model V being 80.1% of that for Model I, whereas R is the worst, with $MSE_{overall}$ for Model
II being 99.3% of that for Model I. For dry land & irrigated land and shrub land, Model II with R factor and
land use as the auxiliary information performed even worse than Model I assisted by the land use. K and L
contributed the similar amount of information for the spatial model, decreasing the MSE about 10% comparing
with Model I. Model VI with R, K and land use as the auxiliary information is superior to any model with land
use and single erosion factor as the covariates (Models I-V). When L and S factor were derived from 30-m
DEM or 90-m DEM, the MSEs are much greater than Model I, which suggested the topography factors help the
interpolation only if the resolution of DEM used to generate them is high enough, such as 1:10000 topography
map. The use of factors derived from DEM with a resolution equal or lower than 30-m seriously worsen the
estimation.
Table 2 summarized the MEs for different land uses and overall data based on different models. All MEs were
greater than 0, except four cases for the Paddy land, which may be due to the limited sample size. Shrub land
and Grassland were the best estimated land use for Model I-VI. All seven models had the overall ME no less
than 0.55, with Model VII having the highest (0.75). The improvements of Model VII comparing with the other
six models were obvious for most land uses. Fig. 6 showed the comparison of predicted and observed soil loss
based on Model VII for four main land uses including dry land & irrigated land, forest, shrub land and
grassland, occupying 30.2%, 15.9%, 7.2%, 37.7% of the total area for Shaanxi province, respectively. It also
showed the predictions of soil erosion on the shrub land and grassland were superior to those on the dry land &
irrigated land and forest, the latter of which existed a degree of underestimation for larger soil loss values (Fig.

23 6).

**3.2 Soil erosion intensity levels and soil loss rates for different land uses**
Models IV, V, and VII require the high resolution of topography maps to derive L and S factor, which we can't
afford in this study; therefore, four soil loss maps based on Models I, II, III and VI were generated. The
proportion pattern of soil erosion intensity levels for all land uses (Fig. 7) and that for different land use (Fig. 8)
were very similar among four models.
The result of Model VI with BPST method showed that the highest percentage is the mild erosion (45.7%),
followed by the slight (20.7%), moderate (19.7%) and high erosion (8.0%). The severe and extreme erosion
were 5.5% and 0.4%, respectively (Fig. 7). When it came to the land use (Fig. 8), the largest percentage for the
dry land & irrigated land was the high erosion, which occupied 23.2% of the total dry land & irrigated land. The
severe and extreme erosion for the dry land & irrigated land were 18.3% and 1.3%, respectively. The largest
percentage for the forest land and grassland was the mild erosion, being 75.1% and 41.7%, respectively. The
percentage of the mild, slight and moderate erosion for the shrub land occupied about 30%, respectively.
Fig. 9 showed soil loss rates for the four main land uses generated from four models. Similar to the estimation of
soil erosion intensity levels, there were slight differences among four models. The soil loss rates for four main
land uses (dry land & irrigated land, forest, shrub land and grassland) by Model VI were reported in Table 3.
**3.3 Spatial distribution of soil erosion intensity**
All four models predicted generally similar spatial patterns of soil erosion intensity, with the mild, moderate and
high erosion mainly occurring in the farmlands and grassland in the northern Loess Plateau region and severe
and extreme soil erosion mainly occurring in the farmlands in the southern Qingba mountainous area (Fig. 10
(a)-(d)). The estimation from Model VI showed that annual soil loss from Shaanxi province was about 207.3 Mt,
49.2% of which came from dry and irrigated lands and 35.2% from grasslands (Table 4). The soil loss rate in
Yan'an and Yulin in the northern part was 16.4 and 13.4 t ha$^{-1}$ y$^{-1}$ and ranked the highest among ten prefecture
cities. More than half of the soil loss for the entire province was from these two districts (Table 4). Ankang and
Hanzhong in the southern part also had a severe soil loss rate and contributed nearly one quarter of soil loss for
the entire province. The soil loss rate in Tongchuan in the middle part was 10.2 t ha$^{-1}$ y$^{-1}$, ranking the fourth
severest, whereas the total soil loss amount was 3.9 Mt, ranking last, due to its smallest area.
**4 Discussion**
**4.1 The uncertainty of the assessment**
The uncertainty of the regional soil loss assessment method combining the survey sample and geostatistics
mainly came from the estimation of erosion factors in the PSU, the density of survey sampling and interpolation
methods. Previous studies have shown that the resolution of topography data source largely affected the
calculated slope steepness, length and soil loss. Thomas et al. (2015) showed that the range of LS factor values
derived from four sources of DEM (20 m DEM generated from 1:50,000 topographic maps, 30-m DEM from
Advanced Spaceborne Thermal Emission and Reflection Radiometer (ASTER), 90-m DEM from SRTM and
250 m DEM from global multi-resolution terrain elevation data (GMTED)) were considerably different, which
suggested the grid resolutions of factor layers are critical and determined by the data resolution used to derive
the factor. Wang et al. (2016) compared data sources including topographic maps at 1:2000, 1:10,000, and
1:50,000 scales, and 30-m DEM from ASTER V1 dataset and reported slope steepness generated from the 30-m
ASTER dataset was 64 % lower than the reference value generated from the 1:2000 topography map (2-m grid)
for a mountainous watershed. The slope length was increased by 265% and soil loss decreased by 47%
compared with the reference values. A study conducted by our research group indicated L and S factor and the
soil loss prediction based on the DEM grid size less than or equal to 10 m were close to those of 2-m DEM (Fu
et al., 2015), therefore, topography maps with a scale of 1:10000 were collected in this study to derive LS-factor
for the PSU. Note that R and K factors for PSUs were clipped from the map of the entire country, which may
include some errors comparing with those from at-site rainfall observation and soil field sampling for each PSU,
which requires further research.
The density of sample units in our survey depends on the level of uncertainty and the budget of the survey. We
tested sample density of 4% in four experimental counties in different regions over China and found a density of
1% was acceptable given the current financial condition. Since our data are a little sparse in some areas, we
employed the roughness penalties to regularize the spline fit; see the energy functional defined in equation (7).
When the sampling is sparse in a certain area, the direct BPST method may not be effective since the results
may have high variability due to the small sample size. The penalized BPST is more suitable for this type of
data because the penalty regularizes the fit (Lai and Wang, 2013).
Cross-validation in section 3.1 evaluated the uncertainty in the interpolation. The results consolidated the
conclusion on the importance of topography factors and the DEM resolution used to calculate topography
factors from previous research. It clarified S factor is the most important auxiliary factor in terms of the
covariate in the interpolation of soil loss and K factor and L factor ranked the second most important, when
topography factors were generated from 1:10000 map. Inclusion of topography factors from 30-m or coarser
resolution of DEM data worsened the estimation.
**4.2 Comparison with the other assessments**
The Ministry of Water Resources of the People's Republic of China (MWR) has organized four nationwide soil
erosion investigations. The first three (in mid-1980s, 1999 and 2000) were mainly based on field survey, visual

interpretation by experts and factorial scoring method (Wang et al., 2016). The third investigation used 30-m

resolution of Landsat TM images and 1:50000 topography map. Six soil erosion intensities were classified

mainly based on the slope for the arable land and a combination of slope and vegetation coverage for the non-

arable land. The limitations for the first three investigations include the limited resolution of satellite images and

topography maps, limited soil erosion factors considered (rainfall erosivity factor, soil erodibility factor, and

practice factor were not considered), incapability of generating the soil erosion rate, and incapability of

assessing the benefit from the soil and water conservation practices. The spatial pattern of soil erosion in

Shaanxi province in this study is similar to the result of the third national investigation. Since the expert factorial

scoring method did not generate the erosion rate for each land use, we compared the percentage of soil erosion

area for ten prefecture cities in Shaanxi province with the third and the fourth investigations. Both investigations

indicated Yan'an, Yulin in the northen part, Tongchuan in the middle part and Ankang in the southern part had

the most serious soil erosion. The difference is that Hanzhong was underestimated and Shangluo was

overestimated in the third investigation, compared with the fourth investigation.

Guo et al. (2015) analyzed 2823 plot-year runoff and soil loss data from runoff plots across five water erosion

regions in China and compared the results with previous research around the world. The results convey that

there were no significant differences for the soil loss rates of forest, shrub land and grassland worldwide,

whereas the soil loss rates of farmland with conventional tillage in northwest and southwest China were much

higher than those in most other countries. Shaanxi province is located in the Northwest (NW) region. Soil loss

rates for the farmland, forest, shrub land and grassland based on the plot data for the NW region in Guo et al.

(2015) were extracted and presented in Table 3 for comparison. Soil loss rate for the farmland based on the plot

data varied greatly with the management and conservation practices and the result in this study was within the

range (Table 3). The soil loss rate for the shrub land is similar with that reported in Guo et al. (2015). The soil

loss rate for the forest in this study was 3.51 t ha$^{-1}$ y$^{-1}$ with a standard deviation of 2.77 t ha$^{-1}$ y$^{-1}$ , which is much

higher than 0.10 t ha$^{-1}$ y$^{-1}$ reported in Guo et al. (2015, Table 3). Our analysis proves that it came from the

estimation of PSUs and was not introduced by the spatial interpolation process. Possible reasons include: (1) the

different definitions of forest and grassland; (2) concentrated storms with intense rainfall; (3) the unique

topography in Loess plateau and (4) the sparse vegetation cover due to intensive human activities (Zheng and

Wang, 2014). The minimum canopy density (crown cover) threshold for the forest across the world vary from

10-30% (Lambrechts et al., 2009) and a threshold of 10% was used in this study, which suggests on average a

lower cover coverage and higher B factor. Annual average precipitation varies between 328-1280 mm in

Shaanxi, with 64% concentrating in June through September. Most rainfall comes from heavy storms of short
duration, which suggests the erosivity density (rainfall erosivity per unit rainfall amount) is high. The field
survey result on the PSUs in this study discovered that the slope degree is steeper and slope length is longer for
the forest than the forest plots in Guo et al. (2015). The forest plots in Guo et al. (2015) were with an averaged
slope degree of 25.9 ° and slope length of 21.1 m, whereas 74.0% of forest lands were with a slope degree
greater than 25 ° and 97.2% of them with a slope length longer than 20 m. The runoff and sediment discharge
observation information for two watersheds (Fig. 1, Table 5) depicted that the soil loss rate for the forest in the
study area has large variability ranging from 1.3 to 19.0 t ha$^{-1}$ y$^{-1}$ (Wang and Fan, 2002). Our estimation is
within the range. The soil loss rate for the grassland in this study was 7.27 t ha$^{-1}$ y$^{-1}$, which was smaller than
11.57 t ha$^{-1}$ y$^{-1}$ reported in Guo et al. (2015). The reason may be due to the lower slope degree for the grassland
in Shaanxi province. The mean value of the slope degree for grassland plots was 30.7 ° in Guo et al. (2015),
whereas 68.6% of the grass lands were with a slope degree smaller than 30 ° from the survey in this study.
Raster multiplication is a popular model-based approach due to its lower cost, simpler procedures and easier
explanation of resulting map. If the resolution of input data for the entire region is enough to derive all the
erosion factors, raster multiplication approach is the best choice. However, there are several concerns about
raster multiplication approach for two reasons: (1) The information for the support practices factor (P) in the
USLE was not easy to collect given the common image resolution and was not included in some assessments
(Lu et al., 2001; Rao et al., 2015), in which the resulting maps don't reflect the condition of soil loss but the risk
of soil loss. Without the information of P factor, it is impossible to assess the benefit from the soil and water
conservation practices (Liu et al., 2013). (2) The accuracy of soil erosion estimation for each cell is of concern if
the resolution of database used to derive the erosion factors is limited. For example, The LS-factor in the new
assessment of soil loss by water erosion in Europe (Panagos et al., 2015b) was calculated using the 25-m DEM,
which may result in some errors for the mountainous and hilly areas due to the limited resolution of DEM data
for each cell (Wang et al., 2016). In this study, the information we can get at this stage for the entire region is
land use, rainfall erosivity (R) and soil erodibility (K). The other factors were not available or without enough
resolution. It is not difficult to conduct raster layer multiplication technically, however, we think the
multiplication of R and K factors (assuming L=1, S=1, B=1, E=1, T=1) reflects the potential of soil erosion,
which is different from the soil loss estimated in this study. Therefore, we did not compare our method with
raster layer multiplication method. Our recommended approach uses all the factor information that are available
in the entire region (land use, rainfall, soils), and uses spatial interpolation to impute other factor information

which are only available at the sampled PSU (slope degree, slope length, practice and management, aggregated as Q) to the entire region. The rationale behind this approach is to exploit the spatial dependence among these factors to come up with better regional estimates. Since the reality in many countries is that we cannot have all factors measured in all areas in the foreseeable future, or the resolution of data for deriving the factors is limited, we believe our approach provides a viable alternative which is of practical importance.

**4.3 Practical implications**

Remarkable spatial heterogeneity of soil erosion intensity was observed in the Shaanxi province. The Loess Plateau region is one of the most severe soil erosion regions in the world due to seasonally concentrated and high intensity rainfall, high erodibility of loess soil, highly dissected landscape, and long-term intensive human activities (Zheng and Wang, 2014). Most of the sediment load in the Yellow River is originated and transported from the Loess Plateau. Recently, the sediment load of the Yellow River declined to about 0.3 billion tons per year from 1.6 billion tons per year in the 1970s, thanks to the soil and water conservation practices taken in the Loess Plateau region (He, 2016). However, more efforts on controlling human accelerated soil erosion in the farmlands and grasslands are still needed. Soil erosion in southern Qingba mountainous region is also very serious, which may be due to the intensive rainfall, farming in the steep slopes and deforestation (Xi et al., 1997). According to the survey in Shaanxi province, 11.1% of the farmlands with a slope degree ranging 15-25 ° and 6.3% of them greater than 25 °were without any conservation practices. Mountainous areas with a slope steeper than 25 °need to be sealed off for afforestation (grass) without the disturbance of the human and livestock. For those farmlands with a slope degree lower than 25 °, terracing and tillage practices are suggested, which can greatly reduce the soil loss rate (Guo et al., 2015, Table 3). The survey result determined that there were 26.5% of grasslands with a slope degree of 15-25 °and 57.6% of them steeper than 25 °without any conservation practices. Enclosure and grazing prohibition are suggested on the grasslands with steep slope and low vegetation coverage.

Note that CSLE as well as other USLE-based models only simulate sheet and rill erosion, and erosion from gullies is not taken into consideration in this study. Erosion from gullies is also very serious in the Loess Plateau area, and there were more than 140,000 gullies with length longer than 500 m in Shaanxi province (Liu, 2013).

**5 Conclusions**

This regional soil erosion assessment focused on the extent, intensity, and distribution of soil erosion on a

regional scale and it provides valuable information for stakeholders to take proper conservation measures in
erosion areas. Shaanxi province is one of the most severe soil erosion regions in China. A field survey in 3116
PSUs in the Shaanxi province and its surrounding areas were conducted, and the soil loss rates for each land use
in the PSU were estimated from an empirical soil loss model (CSLE). Seven spatial interpolation models based
on BPST method were compared which generate regional soil erosion assessment from the PSUs. Following are
our conclusions:
(1) Slope steepness (S) factor derived from 1:10000 topography map is the best single covariate. The MSE of
the soil loss estimator using the model with the land use and S factor is 20% less than those using the model
assisted by the land use alone. Soil erodibility (K) and slope length (L) information reduce about 10% of the
MSE, respectively. Contribution of rainfall erosivity (R) to the decrease of MSE is less than 1%.
(2)   Model VII with the land use and R, K, L, S as the auxiliary information has the model efficiency of 0.75,
and is superior to any model with land use and single or two erosion factors as the covariates (Model I-VI),
which has the model efficiency varying from 0.55 to 0.64.
(3) The LS-factor derived from 30-m DEM or 90-m DEM is not useful when they were used as the covariates
together with the land use, R and K, with the MSEs increased about two times compared with those for the
model assisted by the land use alone.
(4) Four models assisted by the land use (Model I), the land use and R factor (Model II), the land use and K
factor (Model III), the land use, R and K factor (Model VI) provided similar estimates for proportions in each
soil erosion intensity levels, soil loss rates for different land uses and spatial distribution of soil erosion intensity.
(5) There is 54.3% of total land in Shaanxi province with annual soil loss rate no less than 5 t ha$^{-1}$ y$^{-1}$, and total
annual soil loss amount is about 207.3 Mt. Most soil loss originated from the farmlands and grass lands in
Yan'an and Yulin districts in the northern Loess Plateau region, and Ankang and Hanzhong districts in the
southern Qingba mountainous region. Special attention should be given to the 0.11 million km$^2$ of lands with
soil loss rate equal to or greater than 5 t ha$^{-1}$ y$^{-1}$, especially 0.03 million km$^2$ of farmlands with severe and
extreme erosion (greater than 20 t ha$^{-1}$ y$^{-1}$).
(6) A new model-based regional soil erosion assessment method was proposed, which is valuable when input
data used to derive soil erosion factors is not available for the entire region, or the resolution is not adequate.
When the resolution of input datasets was not adequate to derive reliable erosion factor layers and the budget is
limited, our suggestion is sampling a certain amount of small watersheds as primary sampling units and putting
the limited money into these sampling units to ensure the accuracy of soil erosion estimation in these units.
Limited money could be used to collect high resolution data such as satellite images and topography maps and
conduct field research to collect information such as conservation practices for these small watersheds. Then we
can use the best available raster layers for land use, R, and K factor for the entire region, construct spatial
models to exploit the spatial dependence among the other factors, and combine them to generate better regional
estimates. The information collected in the survey and the generated soil erosion degree map (such as Fig. 10d)
can help policy-makers to take suitable erosion control measures in the severely affected areas. Moreover,
climate and management scenarios could be developed based on the database collected in the survey process to
help policy-makers in decision making for managing soil erosion risks.
**Acknowledgments**
This work was supported by the National Natural Science Foundation of China (No. 41301281) and the China
Scholarship Council.

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

**Tables**
**Table 1. Mean squared error of soil loss (A) using bivariate penalized spline over triangulation (BPST) per land use**
[1]

| Model[2] | Land use and sample size | | | | | | | | Overall |
|---|---|---|---|---|---|---|---|---|---|
| | Paddy | Dry land & irrigated land | Orchard & garden | Forest | Shrub land | Grass land | Construction land | Bare land | |
| | 82[3] | 1048[3] | 436[3] | 1288[3] | 574[3] | 684[3] | 323[3] | 32[3] | 4467 |
| I | 0.1 | 513.5 | 181.5 | 25.6 | 46.6 | 19.8 | 1.4 | 4623.1 | 187.8 |
| II | 0.0 | 518.5 | 181.4 | 25.5 | 46.7 | 19.5 | 1.4 | 4283.3 | 186.5 |
| III | 0.1 | 461.7 | 175.8 | 24.3 | 38.7 | 17.2 | 1.4 | 3854.5 | 167.8 |
| IV | 0.0 | 458.7 | 164.3 | 24.5 | 40.2 | 15.6 | 1.3 | 4381.3 | 169.8 |
| V | 0.1 | 424.3 | 148.2 | 24.5 | 41.1 | 15.2 | 1.1 | 3033.0 | 150.5 |
| VI | 0.1 | 464.0 | 175.9 | 24.1 | 37.8 | 16.6 | 1.4 | 3495.1 | 165.5 |
| VII (1:10000 map) | 0.0 | 331.7 | 140.8 | 24.1 | 28.5 | 10.3 | 0.9 | 143.1 | 104.8 |
| VII (30-m DEM) | 0.2 | 1155.8 | 309.1 | 94.2 | 510.3 | 331.6 | 1.3 | 12319.3 | 533.2 |
| VII (90-m DEM) | 0.1 | 1309.4 | 239.5 | 81.0 | 317.1 | 227.0 | 1.5 | 15341.0 | 539.4 |

[1] Since Figure 6 showed no obvious systematic bias for four main land uses, we didn't list the bias
separately in this table.
[2] Model I: Estimating A with the land use as the auxiliary information ; Model II: Land use and R factor as
auxiliary information; Model III: Land use and K factor as auxiliary information; Model IV: Land use and L
factor as auxiliary information; Model V: Land use and S factor as auxiliary information; Model VI: Land use, R
and K factors as auxiliary information; Model VII (1:10000 map) : Land use, R, K, L and S factors as auxiliary
information, and the L factor and S factor were derived from 1:10000 topography maps for the PSUs; Model VII
(30-m DEM): Land use, R, K, L and S factors as auxiliary information, and the L factor and S factor were
derived from 30-m SRTM DEM data for the PSUs; Model VII (90-m DEM): Land use, R, K, L and S factors as
auxiliary information, and the L factor and S factor were derived from 90-m SRTM DEM data for the PSUs.
[3] Sample size for each land use.
**Table 2. Model efficiency coefficient (ME) for seven models using bivariate penalized spline over triangulation**
**(BPST) per land use**

| Model | Land use and sample size | | | | | | | | Over all |
|---|---|---|---|---|---|---|---|---|---|
| | Paddy | Dry land & irrigated land | Orchard & garden | Forest | Shrub land | Grass sland | Constructio n land | Bare land | |
| | 82 | 1048 | 436 | 1288 | 574 | 684 | 323 | 32 | 4467 |
| I | -0.68 | 0.34 | 0.23 | 0.20 | 0.60 | 0.52 | 0.06 | 0.18 | 0.55 |
| II | 0.05 | 0.34 | 0.23 | 0.20 | 0.60 | 0.53 | 0.08 | 0.24 | 0.55 |
| III | -1.98 | 0.41 | 0.26 | 0.24 | 0.67 | 0.59 | 0.08 | 0.32 | 0.60 |
| IV | 0.15 | 0.41 | 0.31 | 0.23 | 0.65 | 0.62 | 0.16 | 0.22 | 0.59 |
| V | -0.08 | 0.46 | 0.37 | 0.23 | 0.65 | 0.63 | 0.26 | 0.46 | 0.64 |
| VI | -0.65 | 0.41 | 0.26 | 0.24 | 0.68 | 0.60 | 0.10 | 0.38 | 0.60 |
| VII (1:10000 map) | 0.82 | 0.58 | 0.40 | 0.25 | 0.76 | 0.75 | 0.43 | 0.97 | 0.75 |

2 **Table 3. Soil loss rates (t ha$^{-1}$y$^{-1}$) for the farmland, forest, shrub land and grassland by Model VI in this study and in**

3 **Northwest region of China from Guo et al. (2015).**

| | Land use | Mean | Standard deviation |
|---|---|---|---|
| This study | Dry land & irrigated land | 21.77 | 20.06 |
| | Forest | 3.51 | 2.77 |
| | Shrub land | 10.00 | 7.51 |
| | Grassland | 7.27 | 5.20 |
| Guo et al. (2015) | Farmland (Conventional) | 49.38 | 57.61 |
| | Farmland (Ridge tillage) | 19.27 | 13.35 |
| | Farmland (Terracing) | 0.12 | 0.28 |
| | Forest | 0.10 | 0.12 |
| | Shrub land | 8.06 | 7.47 |
| | Grassland | 11.57 | 12.72 |

**1** **Table 4. Annual soil loss amount, mean rate and main sources by Model VI for ten prefecture cities in Shaanxi**

**2** **province.**

| Prefecture city | Area ($10^4$ ha) | Amount ($10^6$ t $y^{-1}$) | Mean rate (t $ha^{-1}$ $y^{-1}$) | Source (%) | | | |
|---|---|---|---|---|---|---|---|
| | | | | Dry land and irrigated land | Forest | Shrub land | Grass land |
| Xi'an | 100.9 | 6.5 | 6.4 | 55.0 | 11.2 | 7.8 | 19.6 |
| Ankang | 234.1 | 27.4 | 11.7 | 46.7 | 9.4 | 2.5 | 38.5 |
| Baoji | 180.1 | 14.8 | 8.2 | 36.4 | 10.8 | 7.3 | 39.6 |
| Hanzhong | 268.1 | 20.9 | 7.8 | 45.5 | 11.4 | 3.2 | 36.5 |
| Shangluo | 194.8 | 5.8 | 3.0 | 38.3 | 19.4 | 8.4 | 27.4 |
| Tongchuan | 38.8 | 3.9 | 10.2 | 40.1 | 7.2 | 23.2 | 28.2 |
| Weinan | 129.8 | 7.5 | 5.7 | 59.6 | 3.2 | 8.8 | 24.6 |
| Xianyang | 102.8 | 5.6 | 5.5 | 46.3 | 3.1 | 3.5 | 14.2 |
| Yan'an | 369.1 | 60.5 | 16.4 | 45.7 | 4.8 | 12.0 | 37.0 |
| Yulin | 422.7 | 56.5 | 13.4 | 56.3 | 2.2 | 3.6 | 36.4 |
| Overall | 2041.4 | 207.3 | 10.2 | 49.2 | 6.7 | 7.1 | 35.2 |

**Table 5. Soil erosion rate for the forest and sediment discharge for two watersheds**

| | Area (10⁴ ha) | Runoff (10⁹ m³ y⁻¹) | Sediment discharge (10⁶ t y⁻¹) | Soil loss rate[3] (t ha⁻¹ y⁻¹) | Percent of forest (%) | Soil loss rate for forest (t ha⁻¹ y⁻¹) |
|---|---|---|---|---|---|---|
| Jinghe[1] | 454.2 | 1.837 | 246.7 | 54.3 | 6.5 | 19.0 |
| Luohe[2] | 284.3 | 0.906 | 82.6 | 29.1 | 38.4 | 1.3~2.1 |

[1] Based on the observation at Zhangjiashan hydrological station from 1950 through 1989;
[2] Based on the observation of at Zhuanghe hydrological station from 1959 through 1989;
[3] Sediment delivery ratio, the ratio of sediment discharge from the watershed outlet to the total soil loss,
was assumed to be one; Soil loss rate was defined as the soil loss per unit area.

1 **Figures**

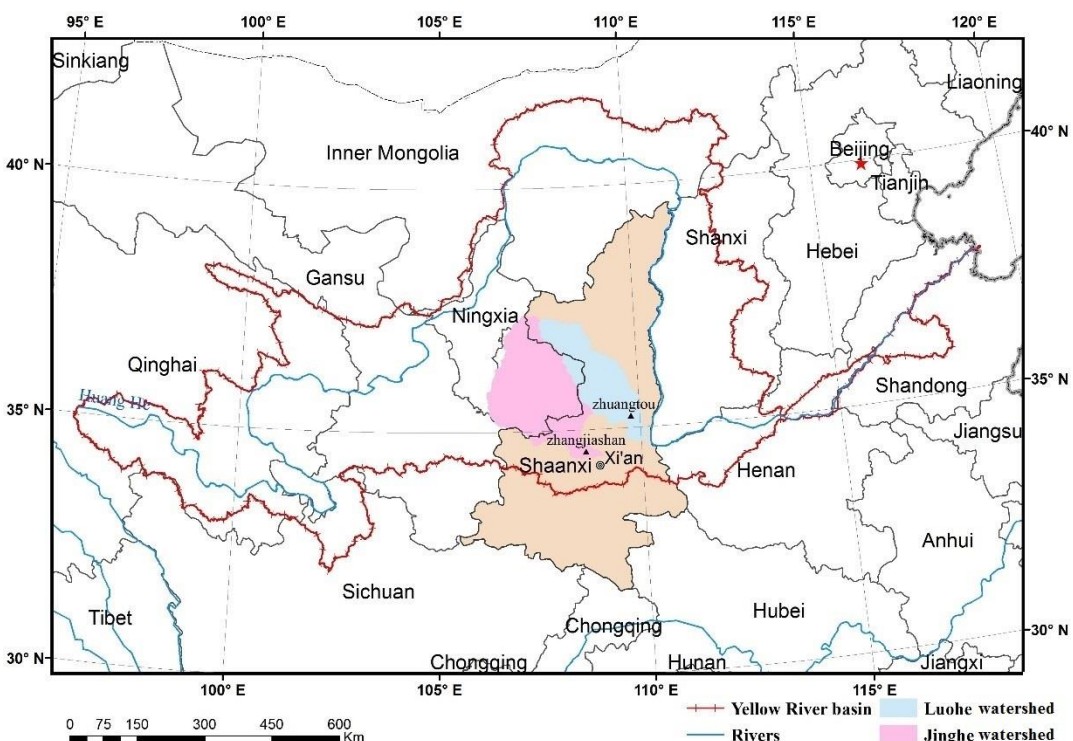

3 **Figure 1: Location of Shaanxi province. Luohe and Jinghe watersheds were referred in the Table 5 and discussion part.**

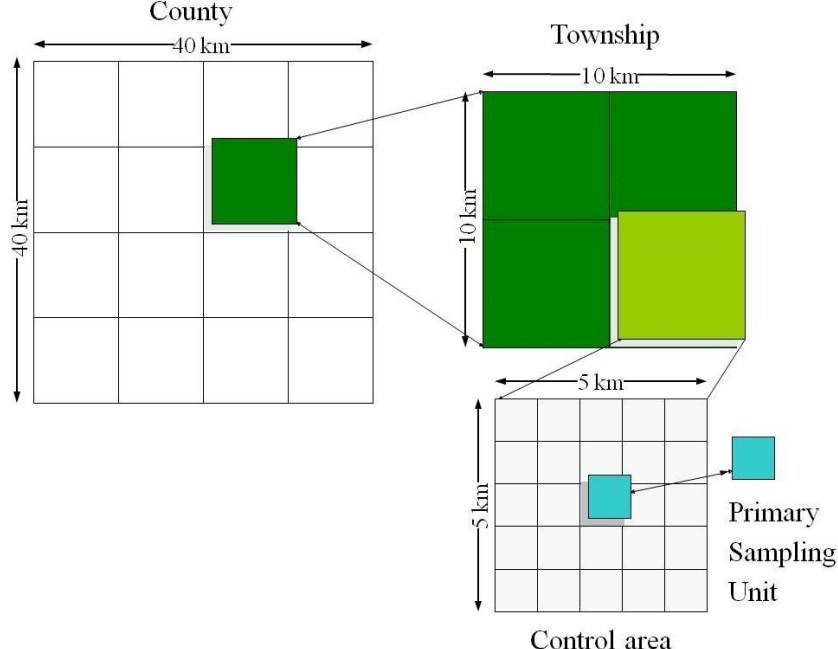

3    **Figure 2: Schematic of sampling strategy for the fourth census on soil erosion in China**

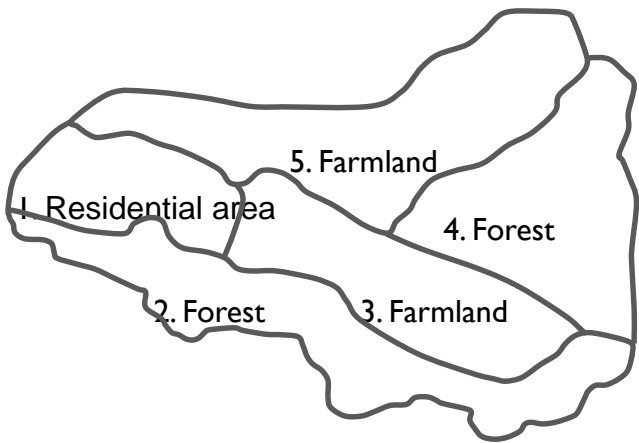

3    **Figure 3: An example of a PSU with five plots and three categories of land uses (Farmland, Forest and Residential**
4    **area).**

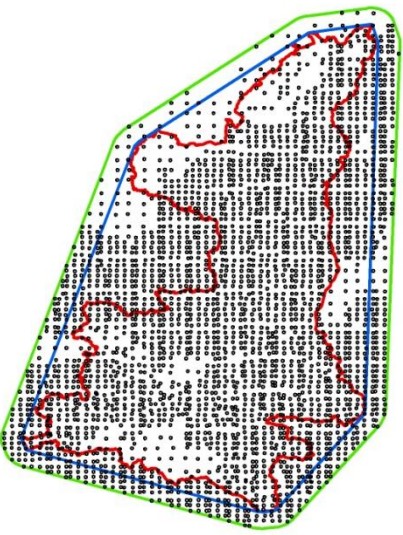

2    **Figure 4: Distribution of PSUs (solid dots) used in this study. The red line is the boundary of the Shaanxi province,**

3    **blue line is the convex hull of the boundary and green line is the 30 km buffer of the convex hull.**

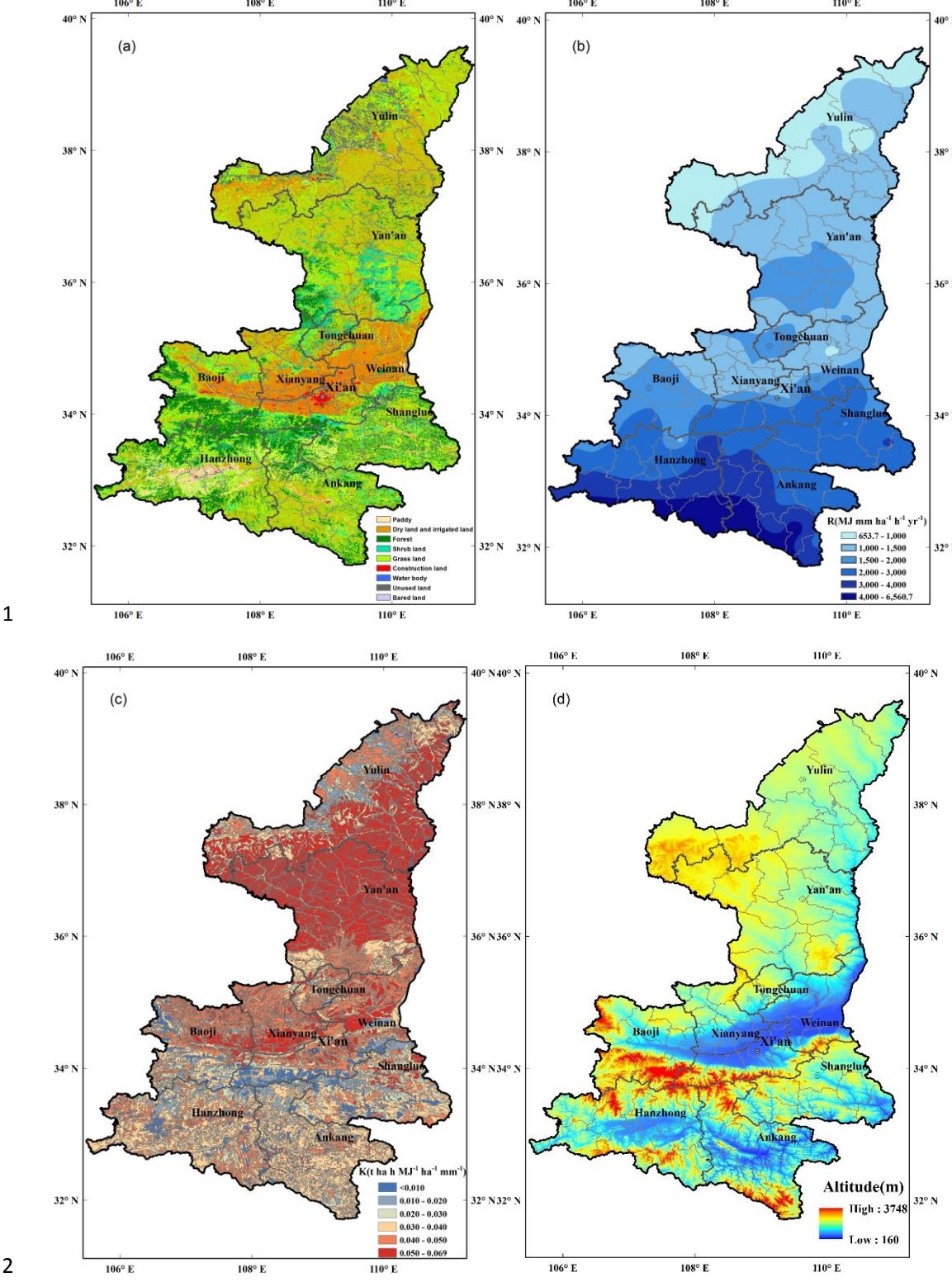

Figure 5 Spatial distributions of land use (a), rainfall erosivity (b), soil erodibility (c) and topography (d) for Shaanxi province.

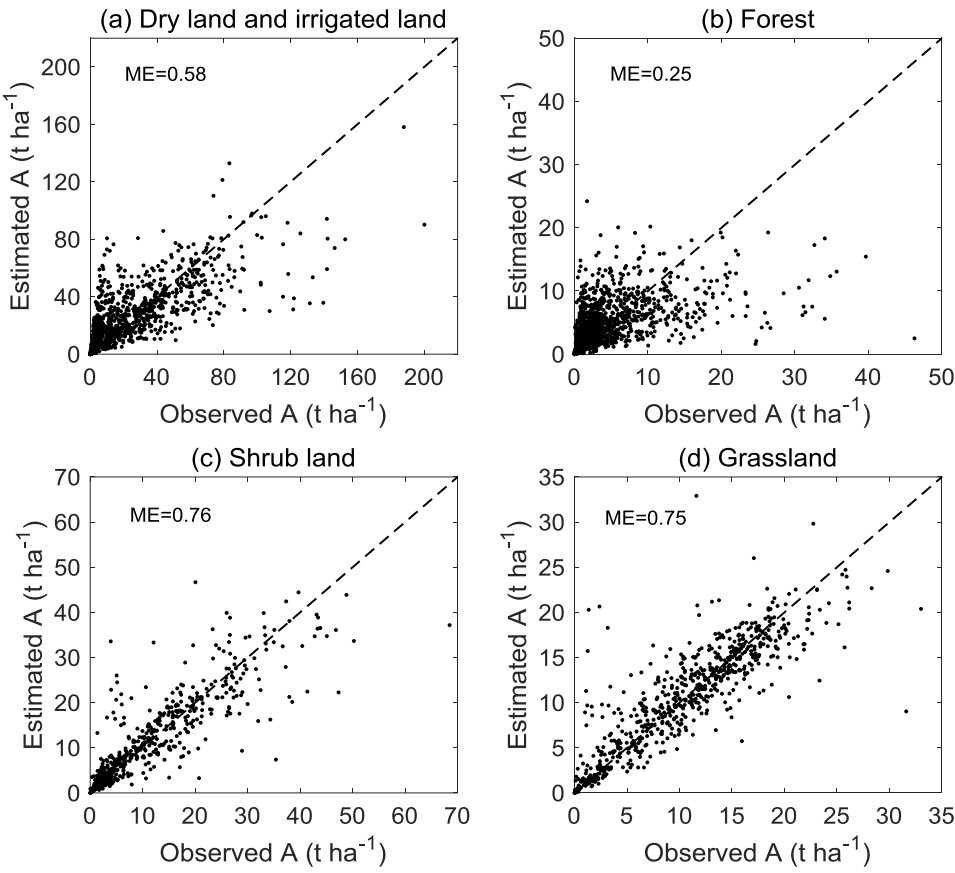

**Figure 6 Scatterplot of estimated and observed soil loss based on Model VII for (a) dry and irrigated land; (b) forest; (c) shrub land; and (d) grassland.**

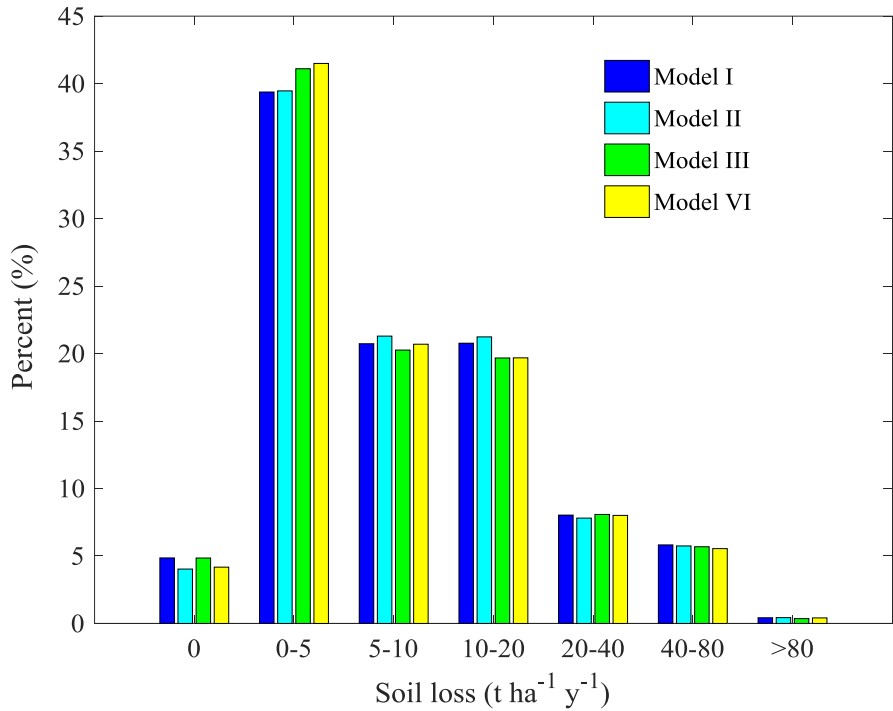

2    **Figure 7: Proportion of soil erosion intensity levels for four models including Model I, II, III and VI.**

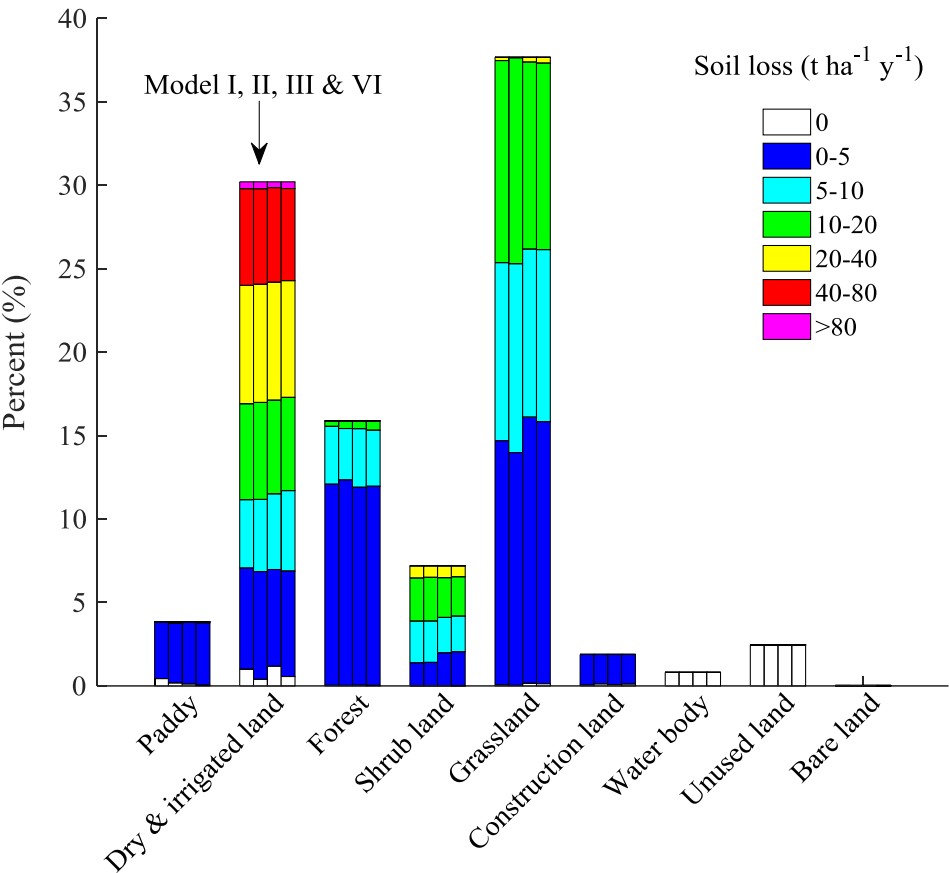

**Figure 8: Proportion of soil erosion intensity levels for different land use for four models including Model I, II, III**

**and VI.**

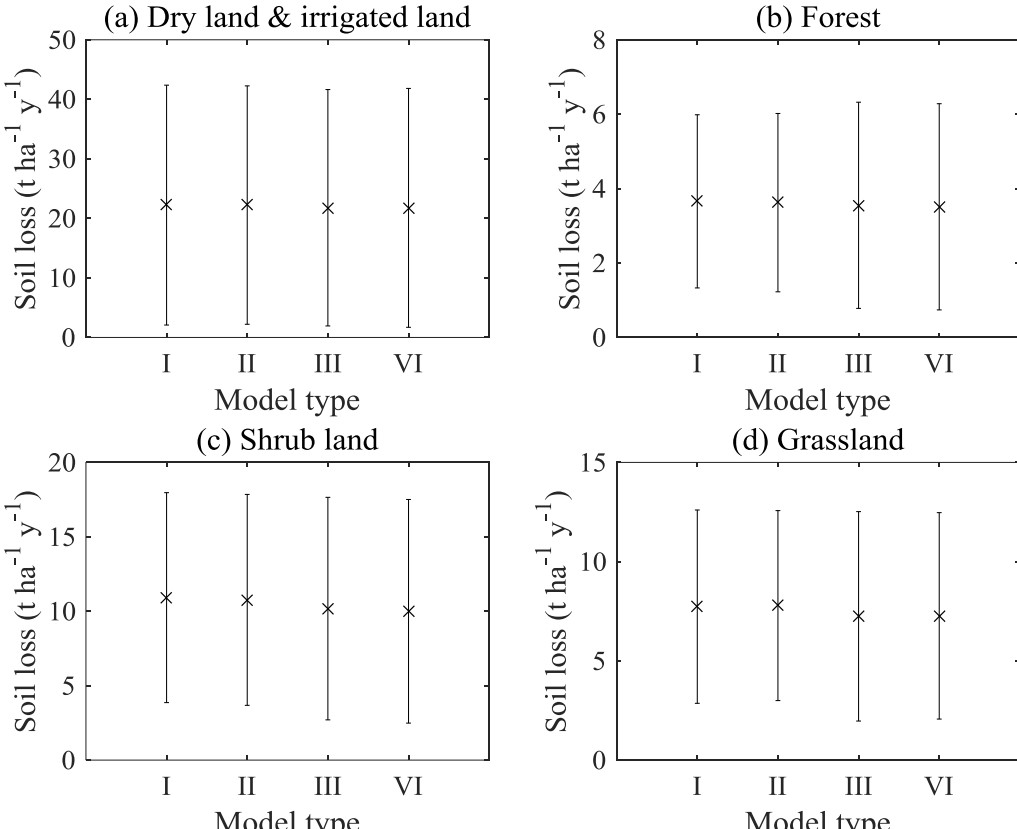

2    **Figure 9: Error bar plot of soil loss rates for four models for different land uses: (a) dry land & irrigated land; (b)**

3    **forest; (c) shrub land; (d) grassland. The star symbols stand for the mean values and the error bars stand for**

4    **standard deviations.**

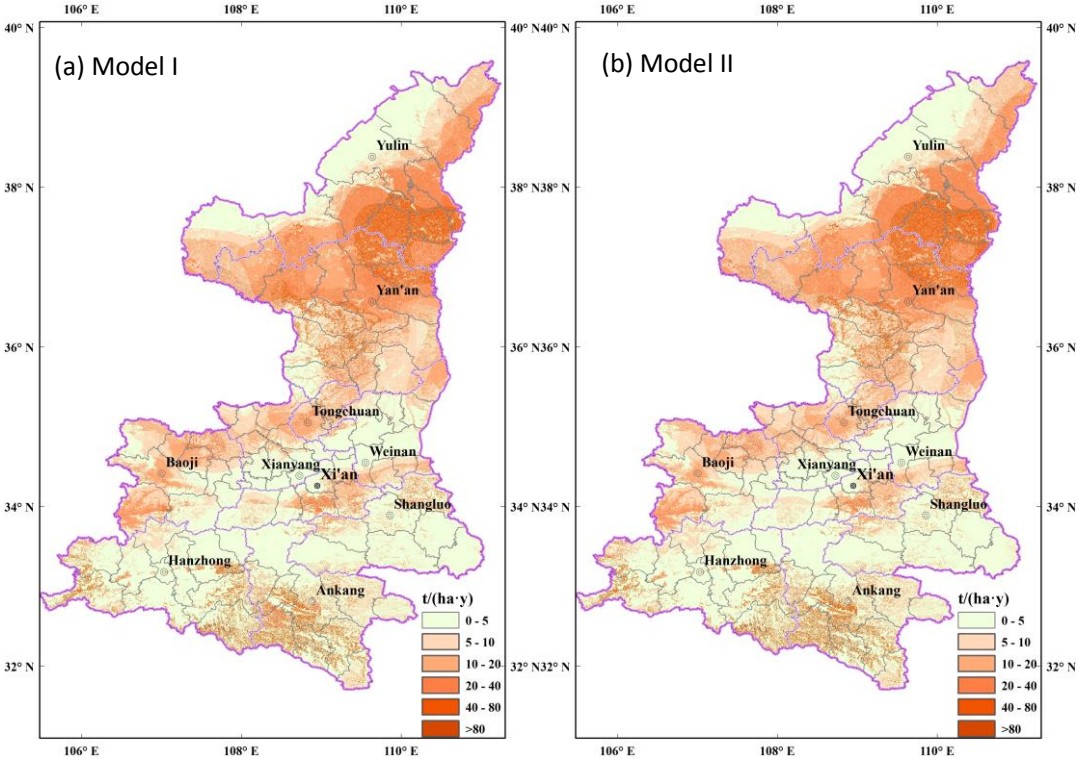

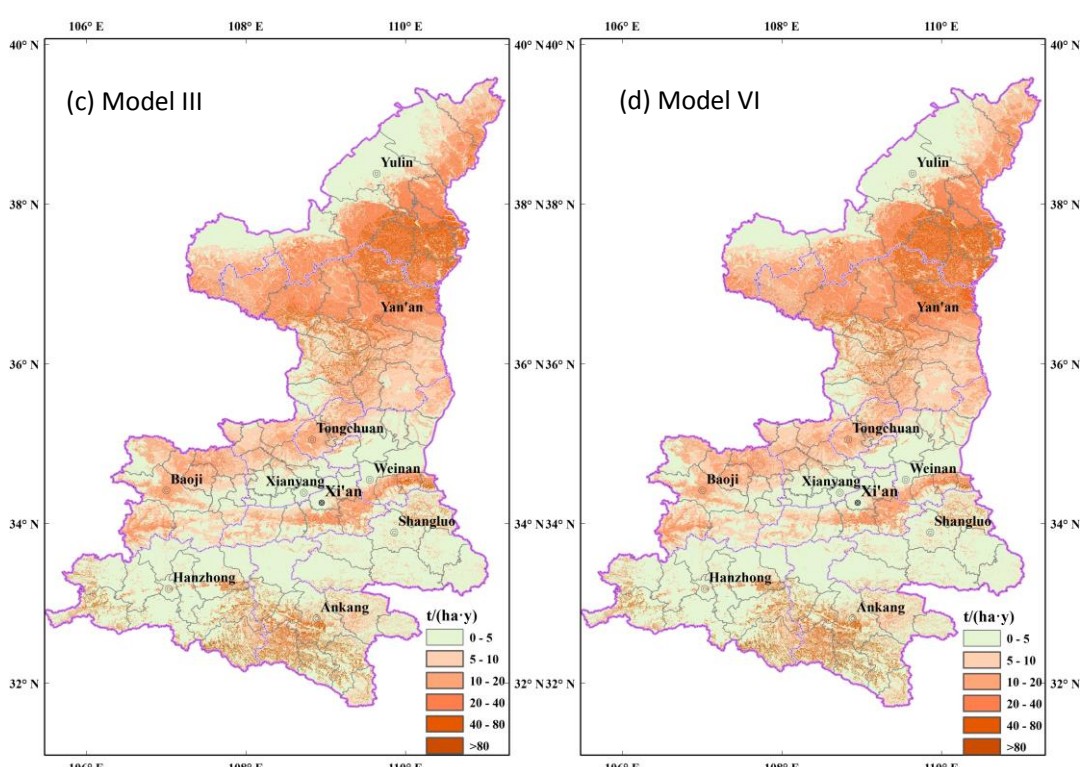

**Figure 10: Distribution of soil erosion intensity levels for four models: (a) Model I; (b) Model II; (c) Model III; (d) Model VI.**