# Peer review of "Regional soil erosion assessment based on sample survey and"

_Hydrology and Earth System Sciences, 2016_

## Referee Comment (RC1) · Anonymous Referee #1 · 29 Sep 2016

Review of the manuscript "Regional soil erosion assessment based on sample survey and geostatistics" by Shuiqing Yin et al.

General comments:

In general, the paper addresses an interesting topic, which would fit into the scope of HESS. Regional soil erosion assessment is still challenging due to the often-missing input data needed for such assessment. Therefore, an alternative approach to the more widespread – mostly USLE based approaches - would be welcome. However, I suggest rejecting the paper for the following reasons: I have general doubts if the produced regional assessment is valuable or if one could learn something regarding the methods presented. The authors use the erosion estimates form 3116 'points' in the Shaanxi province and interpolate these data for a large region using different inter-

polation schemes. Mathematically the interpolation might be correct. However, from an erosion research perspective it just does not make any sense to interpolate erosion data from single locations (with specific land use, slope, slope length, soils, rainfall and soil management) into a large area without taking these important variables into account. The authors present the Chinese variant of the USLE, which identified all important parameters of erosion (Eq. 1, P. 6, line 23), why not using these parameters as co-variables in an interpolation or apply the model itself. From the different interpolation models presented it is obvious that those taking some of the important erosion drivers into account (models II-V) outperform model I, which solely use the erosion data for interpolation. So, concluding my comments I appreciate any efforts to regionalize soil erosion information but I do not think that the presented approach is a promising pathway to follow.

---

## Short Comment (SC1) · 3 Oct 2016

S. Yin

yinshuiqing@bnu.edu.cn

Thank for agreeing with us that the paper addresses an interesting topic and fit into the scope of HESS. The referee mentioned that "Regional soil erosion assessment is still challenging due to the often-missing input data needed for such assessment." which is the main reason why we proposed a method combining the sample survey and geostatistics. We understood the referee had three main concerns:

(1) It just does not make any sense to interpolate erosion data from single locations (with specific land use, slope, slope length, soils, rainfall and soil management) into a large area without taking these important variables into account.

Response: We actually did take these important variables into account to estimate regional soil loss whenever possible. One major point we want to make in the paper is

that the simple interpolation without using any of the available factor information (Model I) is not good. Our recommended approach uses all the factor information that are available in all area (land use, rainfall, soils), and uses spatial interpolation to impute other factor information which are only available at the sampled PSU (slope degree, slope length, practice and management, aggregated as Q) to all area. The rational behind this approach is to exploit the spatial dependence among these factors to come up with better regional estimates. Since the reality in many developing countries is that we cannot have all factors measured in all areas in the foreseeable future, we believe our approach provide a viable alternative which is of practical importance.

(2) The authors present the Chinese variant of the USLE, which identified all important parameters of erosion (Eq. 1, P. 6, line 23), why not using these parameters as co-variables in an interpolation or apply the model itself.

Response: We can only obtain the information for all seven erosion factors in the CSLE in the Primary Sample Unit (PSU), not for the entire domain. For the entire domain, the information we can get in this stage is land use, rainfall erosivity (R) and soil erodibility (K). As explained in (1), we did apply the model itself by using parameters that are available in all area (land use, R and K) as covariate in our semi-parametric model (equation 5, 7 and 9), and interpolate the rest of the parameters aggregated as factor Q. The other factors including slope length, slope degree, biological, engineering and tillage practice factors are impossible or difficult to obtain for the entire domain in this stage. We sampled micro watersheds (PSU) to collect detailed topography information and conducted field survey to collect practice information. Previous research showed that topography factors should be derived from high resolution topography information (such as 1:10000 topography contour map or larger scale). Topography factors based on smaller scale of topography map (such as 1:50000 or 1:100000) in the mountainous area have large uncertainties. We can obtain 1:10000 topography contour map for the PSU, not for the entire domain. For the forest land, the vegetation coverage derived from the remote sensing data was used as the canopy density, which was combined

with the vegetation fraction under the trees collected during the field survey to estimate the half-month biological practice factor. The vegetation fraction under the trees is of great importance in protecting soil and it cannot be derived from remote sensing images. Engineering and tillage practice factors were based on the sample field survey. They are not easy to be interpreted from images with common resolution.

(3) From the different interpolation models presented it is obvious that those taking some of the important erosion drivers into account (models II-V) outperform model I, which solely use the erosion data for interpolation. . . .I do not think that the presented approach is a promising pathway to follow.

Response: Our contribution in this paper is twofold. First, our study quantified how important the knowledge of land use, rainfall erosivity and soil erodibility in all area are for estimating regional soil loss (Land use is the key auxiliary information for the spatial model, which contributed much more information than R and K factors did. See Page 11 Line 8-9). Second, we introduced a new regional soil erosion assessment method, which is different from the area sample survey approach used by NRI in the USA and the multiplication of raster layers used by Europe, Australia, and many other regions (See Page 3 Line 18-30, Page 4 Line 1-13 for the detail). If the resolution of input data for the entire domain is enough to derive all the erosion factors, the multiplication of raster layers is a good choice. However, for many regions in the world, such input data is often not available, or the resolution is not adequate for the assessment, as the referee has mentioned. Therefore, the assessment method combining sample survey and geostatistics proposed in this study is valuable.

---

## Referee Comment (RC2) · Anonymous Referee #2 · 10 Oct 2016

I would disagree with the first review that this is not a valuable paper. The manuscript addresses an interesting issue and if the authors would improve it significantly then it can be worth publishing. However, there are a number of issues that the authors should address and correct.

The authors present in the introduction a number of methodologies for assessing soil erosion: a) fractional scoring b) plot measurements c) field-based approach d) Modelling (RUSLE). Then they analyse more in detail the application of RUSLE as 3 different options: 1) sample survey 2) raster multiplication 3) sample survey and geostatistics. The authors have followed the third option.

Find below the most important remarks and issues that authors should address in their revision: First remark: I would appreciate if the authors have compared their

results with the second option. This would give much more advanced knowledge in the manuscript. You mentioned that you have available K-factor, R-factor maps at 250m resolution plus a land use map at 100000 scale. So, it would have been excellent to compare your results with an estimated Soil loss by water erosion (simply multiplying the above mentioned high resolution grids).

Second: As the 1st reviewer said (and I agree) , the authors have presented an interpolation method which takes into account 5 different group of parameters. It is logical (and obvious that the IV and V would perform much better than the I. In a recent research (to be online soon), we identified cover management factor as the most sensitive for estimating soil loss by water erosion. The manuscript could be even more worthy if the authors have compared their findings with alternative methods (plot measurements, expert knowledge, field-based approach).

Third: The findings regarding the forests are much too high. Erosion of > 3 t ha-1 in forest is not at all acceptable. Even there can be very steep slopes, the forestland experience erosion of much less than 1 t ha-1 annually. Their comparison with the findings of Guo (2015) and the findings in Europe (2015) show that erosion in forests is much less. The same applies for grasslands.

Please consider also a comparison of your findings with the paper of Wang et al (2016) "Assessment of soil erosion change and its relationships with land use/cover change in China from the end of the 1980s to 2010"

Fourth: authors should explain and justify the selection of their statistical model BPST and not the selection of Cubist or GPR or regression kriging? Moreover, In your geo-statistical model , the topography is ignored. Why?

Fifth: The filed survey (section 2.1) indicates that the sampling of erosion points was not so dense. Please give some levels of uncertainty taking into account that you sampled on PSU every 25 km2 even less. Moreover, you mentioned that "PSU points were surveyed" : you don't describe how you estimate the R, K, LS, B, E, T factors

in each point? Did you sample and analyse the soil for estimating K-factor? Did you install a high temporal resolution rainfall station for measuring R-factor? Etc . Maybe this is somehow written in section 2.3 but it is not clear as you don't provide detailed information on how the R-factor, K-factor was calculated.

In the same way that you criticize the non-availability of all input layers when multiplying the grids (factors), somebody can criticize your methodology that non all information (K-factor, R-factor, ect) is available at point level. How you respond to this?

More specific comments related to text: - P2L25-26: Rephrase the sentence. - P2L3: The reference should be Panagos et al, 2016a. The paper of Panagos et al, 2015a in your reference list does not feature in the literature. Please check carefully your literature. The same in P2L6 (it should be Panagos et al 2015; Panagos et al., 2016a). - P3L16-17: You cannot put in the same importance the papers of Bosco (2015) and Panagos (2015) regarding the European soil erosion assessments. The second one is much more advanced with new knowledge. For more info about the model evolution in Europe, please consider the paper of Borrelli et al (2016), Land Use policy. - P3L23-25: some references to NRI methodology and the outcome results are missing here. I would also appreciate some applications and datasets derived (With references) derived from this methodology. - P4L6-9: The European assessments was commented by 2 papers (you have put the 3 comments here) but you ignore the response of Panagos et al (2016a, 2016b) to those 3 critiques. - - Section 2.2 and elsewhere: R-factor, K-factor and land use map. Please give some citations and source of this dada. - You may be familiar with CSLE but for some non-Chinese, it would be better to write 2 lines about the biological and engineering factor. - P7L11-13: Your method has many similarities with the estimation of C-factor in Europe based on vegetation density and land use (Panagos et al, 2015 Land use policy). - P7 Equations 2 and 3 : they seem to be very similar. Which is the difference, please explain. - Model I has no sense as it is obvious to be so poor!!! - Section 2.4.2: you start without any introduction about BPST model. Please add an introductory paragraph - P11L13-15: it is obvious that

model I and II will have no extreme erosion levels (as the land use is ignored). The smoothening effect is obvious! - Conclusions: P14L5-6: it is better somewhere in the introduction.. In general the conclusions should highlight the important findings of this study. - The way forward: The authors should conclude about the usefulness of their methodology. How this can be used? How it can be complementary to the traditional multiplication of grids.

Fig1: The Dark green image Towner-ship cannot be 10 x 40km? Fig2: The land uses are 5 and not 3. Fig 5: It is difficult to see the distinction between 40-80 and 80 categories in the first bar. Fig. 7: Scale bar and overview map is missing . As a non-Chinese, I don't know where this province is located.

Minor comments - P5L7: please replace with "erodibility" - "soil species" is not a mature term - 3.1 : Four or Five models?

---

## Short Comment (SC2) · 17 Oct 2016

Thanks for the valuable comments by the Referee #2. Here we present responses to five main comments.

1.  The authors present in the introduction a number of methodologies for assessing soil erosion: a) factorial scoring b) plot measurements c) field-based approach d) Modelling (RUSLE). Then they analyse more in detail the application of RUSLE as 3 different options: 1) sample survey 2) raster multiplication 3) sample survey and geostatistics. The authors have followed the third option. Find below the most important remarks and issues that authors should address in their revision: First remark: I would appreciate if the authors have compared their results with the second option. This would give much more advanced knowledge in the manuscript. You mentioned that you have available K-factor, R-factor maps at 250m resolution plus a land use map at 100000 scale. So, it would have been excellent to compare your results with an estimated Soil loss by water erosion (simply multiplying the above mentioned high resolution grids).

    **Response:** It is not difficult to conduct raster layer multiplication technically, however, the multiplication of R and K factors (assuming L=1, S=1, B=1, E=1, T=1) reflect the potential of soil erosion, which is different from the soil erosion estimated in this study.

2.  As the 1st reviewer said (and I agree), the authors have presented an interpolation method which takes into account 5 different group of parameters. It is logical (and obvious that the IV and V would perform much better than the I. In a recent research (to be online soon), we identified cover management factor as the most sensitive for estimating soil loss by water erosion. The manuscript could be even more worthy if the authors have compared their findings with alternative methods (plot measurements, expert knowledge, field-based approach).

    **Response:** In the manuscript, we have compared with the plot measurements (Guo et al., 2015). We could at least compare the result with the third national census for soil erosion, which is based on factorial scoring method (Wang et al., 2016). Thanks for your suggestion.

3. The findings regarding the forests are much too high. Erosion of > 3 t ha-1 in forest is not at all acceptable. Even there can be very steep slopes, the forestland experience erosion of much less than 1 t ha-1 annually. Their comparison with the findings of Guo (2015) and the findings in Europe (2015) show that erosion in forests

   is much less. The same applies for grasslands. Please consider also a comparison of your findings with the paper of Wang et al (2016) "Assessment of soil erosion change and its relationships with land use/cover change in China from the end of the 1980s to 2010".

   **Response:** Wang et al. (2016) used a factorial scoring method to assess soil erosion risk and change in China from the end of the 1980s to 2010. As it was discussed in the introduction of this study, the resulting map by the factorial scoring method depicts classes ranging from very low to very high erosion or erosion risk. However, it can't generate erosion rates. We are also concerned about the relatively higher erosion rates of forest and grassland in the Shaanxi province comparing with some previous research. Preliminary analysis showed that they came from the Primary Sampling Units, and not introduced by the spatial interpolation process. The reason for this may be due to the different definitions of forest and grassland, the unique topography in Loess plateau and intensive human activities. The minimum canopy density (crown cover) threshold for the forest in this study is 10%, which may suggest a lower cover coverage and higher B factor. The grassland includes the native and artificial grassland, with more intensive livestock and human activities. But more analysis is required and we will do it before we make the revision.

4. Authors should explain and justify the selection of their statistical model BPST and not the selection of Cubist or GPR or regression kriging? Moreover, In your geo-statistical model, the topography is ignored. Why?

   **Response:** In spatial data analysis, there are mainly two approaches to make the

prediction of a target variable. One approach (e.g., kriging) treats the value of a target variable at each location as a random variable and uses the covariance function between these random variables or a variogram to represent the correlation; another approach (e.g., spline or wavelet smoothing) uses a deterministic smooth surface function to describe the variations and connections among values at different locations. Our work takes the second approach. The relationship between the traditional spatial statistics, and splines have been discussed in the literature, e.g. Matheron (1981) and Wahba (1990). A brief comment is presented in the following. Specifically, as discussed in Mitas and Mitasova (1999), "Kriging assumes that the spatial distribution of a geographical phenomenon can be modeled by a realization of a random function and uses statistical techniques to analyze the data and statistical criteria for predictions. However, subjective decisions are necessary such as judgement about stationarity, choice of function for theoretical variogram, etc. In addition, often the data simply lack information about important features of the modelled phenomenon, such as surface analytical properties or physically acceptable local geometries." In contrast, "Splines rely on a physical model with flexibility provided by change of elastic properties of the interpolation function. Often, physical phenomena result from processes which minimize energy, with a typical example of terrain with its balance between gravitation force, soil cohesion, and impact of climate. For these cases, splines have proven to be rather successful." For our problem, we also pay special attention to the following two practical issues: (1) the data are not necessarily evenly distributed; observations can be dense at some locations while sparse at others. (2) the domain for the data can take non-rectangular shapes.

In this work we introduce bivariate splines on triangulations to handle irregular domains and propose to extend the idea of univariate penalized splines (Eilers and Marx, 1996) to the two-dimensional case. The BPST method we consider have several advantages. First, it provides good approximations of smooth functions over complicated domains. Second, the computational cost for spline

evaluation and parameter estimation are manageable. Third, the BPST doesn't require the data to be evenly distributed or on regular-spaced grid. Topography factors based on smaller scale of topography map in the mountainous area have large uncertainties. A recent research by our group showed that the slope steepness based on the 30 m ASTER GDEM V1 is about 64% lower and the slope length on the other hand was increased by 265%, compared with the reference value based on the topography map with a scale of 1:2000 for a mountainous watershed in Northern China. If larger scale of topography map can be collected and it is not difficult to incorporate topography factors into our model by adding L and S factors in the equations (8) and (9).

References:

Mitas, L. and Mitasova, H. (1999). Spatial interpolation. Geographical information systems: principles, 1, 481-492.

Matheron, G. (1981) Splines and Kriging: their formal equivalence. Syracuse University Geological Contributions: 77–95.

Wahba, G. (1990). Spline models for observational data. CNMS-NSF Regional conference series in applied mathematics 59. Philadelphia, SIAM.

5. The filed survey (section 2.1) indicates that the sampling of erosion points was not so dense. Please give some levels of uncertainty taking into account that you sampled on PSU every 25 km2 even less. Moreover, you mentioned that "PSU points were surveyed" : you don't describe how you estimate the R, K, LS, B, E, T factors in each point? Did you sample and analyse the soil for estimating K-factor? Did you install a high temporal resolution rainfall station for measuring R-factor? Etc. Maybe this is somehow written in section 2.3 but it is not clear as you don't provide detailed information on how the R-factor, K-factor was calculated. In the same way that you criticize the non-availability of all input layers when multiplying the grids (factors), somebody can criticize your methodology that non all information (K-factor, R-factor, ect) is available at point

level. How you respond to this?

**Response:** The density of sample units in our survey depends on the level of uncertainty and the budget of the survey. We sampled a density of 4% in four experimental counties in different regions over China and found a density of 1% was acceptable given the current financial condition.

Lai and Wang (2013) provided the asymptotic properties of the BPST method, For example, they investigated how the bias and variance of the BPST estimator change with respect to the sample size and the number of the triangulations. Since our data are a little sparse in some area, we employed the roughness penalties to regularize the spline fit; see the energy functional defined in equation (12). When the sampling is sparse in certain area, the direct BPST method will not be effective since the results of the smoothing stage may have high variability due to the small sample size. The penalized BPST is more suitable for this type of data because it can help to regularize the fit.

We didn't install a rainfall station or collect soil samples for measuring R or K factor for each PSU. Instead, we collected 2678 weather and hydrologic stations with erosive daily rainfall from 1981 through 2010 and generated the R factor raster map over the entire China (Xie et al., 2016). And for the K factor, soil surface attributes for 7764 soil species from the Second National Soil Survey and more than 950 soil samples newly collected were used to generate the K factor for the entire country (Liu et al., 2013). The R and K factors for each PSU were clipped from the map of the entire country. A topography contour map with a scale of 1:10000 for each PSU was collected to derive the slope length and slope degree and to calculate the slope length factor and slope steepness factor (Fu et al., 2013). The calculation of B, E and T was based on the field survey of each PSU. As we know that R factor in USLE requires the breakpoint rainfall data more than 20 years, which is not feasible if a rainfall station was set up in each PSU. We assumed that the variation of R factor could be captured by more than 2000 stations over China, which were the most stations we could collect at present. Soil maps with scales of 1:500,000 to 1:200,000 (for different provinces) generated

more than 0.18 million polygons of soil attributes over mainland China, which was the most accurate soil information we could collect at present. We assumed the result of the soil survey could be used as the information of K factor in our soil erosion survey. R factor and K factor in the sample point of NRI was also from the interpolation result of weather stations and soil survey map, respectively.

References:

Lai, M. J. and Wang, L. (2013). Bivariate penalized splines for regression. Statistica
Sinica 23 1399–1417.

Xie, Y., Yin, S. Q., Liu, B. Y., Nearing M., and Zhao, Y.: Models for estimating daily rainfall erosivity in China. J Hydrol, 535, 547–558, 2016.

Liu, B.Y., Guo, S. Y., Li, Z. G., Xie, Y., Zhang, K. L., and Liu, X. C.: Sample survey on water erosion in China. Soil and Water Conservation in China, 10, 26–34, 2013 (in Chinese with English abstract).

Fu, S. H., Wu, Z. P., Liu, B. Y., and Cao, L. X.: Comparison of the effects of the different methods for computing the slope length factor at a watershed scale. Int Soil Water Conserv Res, 1(2), 64–71, 2013.

---

## Author Comment (AC2) · 17 Nov 2017

Could you please update the status of manuscript hess-2016-394 if possible? Thank you for your time.
* * *

---

## Author Response (AR1)

We would like to thank the editor and all referees for their valuable comments. Moreover, we appreciate that the editor and all referees think the paper addresses an interesting topic and fit into the scope of HESS. We revised the manuscript thoroughly considering all the comments from the editor and referees. Here is a detailed author response to all comments from the editor and referees. The page and line information in the response please refer to the CLEAN VERSION in the supplement.

8

**9 AUTHOR RESPONSE TO THE EDITOR**

**10 General comments:**

1. I agree with the authors and the reviewers, that the manuscript addresses an 11 important topic: estimation of soil erosion rates in a large scale. As acknowledged 12 by all, this task will lead to a number of methodological challenges. The authors 13 14 try to address those challenges by applying a spline-based interpolation technique, using different levels of additional external information, such as land use, rainfall, 15 soil characteristics. I feel that this can be a valuable approach in addressing the 16 challenge of large scale erosion (maps). However, I think the authors do not show 17 (enough) how superior their method is, compared to other techniques. Therefore, I 18 suggest that the authors try to work along the remarks of the reviewers. I also 19 20 agree with rev. #2 that one should compare with other independent soil erosion values, see also below. 21 **Response:** All the remarks of the reviewers were considered in the revision, see 22 the detailed response to comments from the reviewers. Besides the comparison 23 with the plot data in Guo et al., 2015, in the revised version, the result of this 24 study based on the survey data from the fourth national soil erosion investigation 25 26 was compared with the result of the third national investigation (Page 15 Line 19-27 25), and the observations on the soil erosion rate in the forest and sediment discharge from two hydrological stations for two watersheds (Page 16 Line 20-28 22). 29

- I addition, I think that information about the values and the spatial distribution
   (maps) of the additional information (land use, rainfall, soil characteristics,
- topography) should be displayed. This needs to be given in the same spatial detail
  as the interpolation method uses this information.
- Response: Land use, rainfall erosivity, soil erodibility and topography maps have
  been added (Fig. 5).
- 36 3. I also wonder if the authors can give a measured/observed value of a lumped total
  arosion rate in a meso-scale region over some years (e.g. soil deposits in a
  reservoir) and compare such value(s) with their estimates. This would be
- 39 particular helpful to get more confidence into the very high erosion rates in the

1 forest.

2 **Response:** Good suggestion! The sediment discharge observation from two

3 hydrological stations and soil erosion rate for the forest land for two watersheds

4 have been added and the result showed the estimated erosion rate for the forest in

5 this study was consistent with the observation (Page 16 Line 20-22).

6

**7 **Other comment:**

Please do not display internationally disputed borders, as you do in your Fig. 1a,
page 12, in your "AC1: Final response to the comments from two referees". This is
not acceptable for publication in a final version.

11

**Response:** Revised (Fig. 1).

**12**

13

**14 AUTHOR RESPONSE TO RC #1**

**15 General comments:**

In general, the paper addresses an interesting topic, which would fit into the 16 scope of HESS. Regional soil erosion assessment is still challenging due to the often-17 missing input data needed for such assessment. Therefore, an alternative approach to 18 19 the more widespread – mostly USLE based approaches - would be welcome. 20 However, I suggest rejecting the paper for the following reasons: I have general doubts if the produced regional assessment is valuable or if one could learn something 21 regarding the methods presented. The authors use the erosion estimates form 3116 22 'points' in the Shaanxi province and interpolate these data for a large region using 23 different interpolation schemes. Mathematically the interpolation might be correct. 24 However, from an erosion research perspective it just does not make any sense to 25 26 interpolate erosion data from single locations (with specific land use, slope, slope length, soils, rainfall and soil management) into a large area without taking these 27 important variables into account. The authors present the Chinese variant of the 28 USLE, which identified all important parameters of erosion (Eq. 1, P. 6, line 23), why 29 not using these parameters as co-variables in an interpolation or apply the model 30 itself. From the different interpolation models presented it is obvious that those taking 31 some of the important erosion drivers into account (models II-V) outperform model I, 32 which solely use the erosion data for interpolation. So, concluding my comments I 33 34 appreciate any efforts to regionalize soil erosion information but I do not think that the presented approach is a promising pathway to follow. 35 36 We understood the referee had three main concerns: 37

- 38 (1) It just does not make any sense to interpolate erosion data from single
- 39 *locations (with specific land use, slope, slope length, soils, rainfall and soil*
- 40 *management*) into a large area without taking these important variables into

**1 *account*.**

**Response:** We actually did take these important variables into account to estimate 2 regional soil loss whenever possible. One major point we want to make in the 3 paper is that the simple interpolation without using any of the available factor 4 information (Model I) is not good. Our recommended approach uses all the factor 5 6 information that are available in the entire region (land use, rainfall, soils), and 7 uses spatial interpolation to impute other factor information which are only available at the sampled PSU (slope degree, slope length, practice and 8 management, aggregated as Q) to the entire region. The rationale behind this 9 approach is to exploit the spatial dependence among these factors to come up with 10 better regional estimates. Since the reality in many countries is that we cannot 11 have all factors measured in all areas in the foreseeable future, or the resolution of 12 data for deriving the factors is limited, we believe our approach provide a viable 13 alternative which is of practical importance. 14

15 16

17

(2) The authors present the Chinese variant of the USLE, which identified all important parameters of erosion (Eq. 1, P. 6, line 23), why not using these

*parameters as co-variables in an interpolation or apply the model itself.* **Response:** It seems that there is some misunderstanding here. We can only obtain
the information for all seven erosion factors in the CSLE in the Primary Sample
Unit (PSU), not for the entire region. Therefore, it is impossible to using all erosion
parameters as co-variables in an interpolation or conduct a raster multiplication of
all seven parameters in the CSLE. We made this more clear in the revision (Page 6;
Line 3-8; Page 9 Line 7-9 ).

For the entire region, the information we can get at this stage is land use, rainfall erosivity (R) and soil erodibility (K). As explained in (1), we did apply the model itself by using parameters that are available in all area (land use, R and K) as covariates in our semi-parametric model (equation 5, 7 and 9), and interpolate the rest of the parameters aggregated as factor Q.

30 The other factors including the slope length, slope degree, biological, engineering and tillage practice factors are either impossible or very difficult to 31 obtain for the entire region at this stage. We sampled small watersheds (PSUs) to 32 collect detailed topography information and conducted field survey to collect soil 33 34 and water conservation practice information. Previous research showed that topography factors should be derived from high resolution topography information 35 (such as 1:10000 or larger scale topography contour map). Topography factors 36 based on smaller scale of topography map (such as 1:50000 or 1:100000) in the 37 mountainous and hilly area have large uncertainties. We can obtain 1:10000 38 topography contour map for the PSUs, but not for the entire region. For the forest 39 land, the vegetation coverage derived from the remote sensing data was used as the 40 canopy cover density, which was combined with the vegetation fraction and residue 41 42 under the trees collected **during the field survey** to estimate the half-month biological practice factor. The vegetation fraction and residue under the trees is of 43 great importance in protecting soil and it cannot be derived from satellite images. 44

Engineering and tillage practice factors were based on the sample field survey. It
 is difficult to collect these factors from images with common resolution.

3

(3) From the different interpolation models presented it is obvious that those 4 taking some of the important erosion drivers into account (models II-V) 5 6 outperform model I, which solely use the erosion data for interpolation. ... I do not think that the presented approach is a promising pathway to follow. 7 Response: Our contribution in this paper is twofold. First, our study quantified how 8 important the knowledge of land use, rainfall erosivity and soil erodibility in all 9 area are for estimating regional soil loss (Land use is the key auxiliary information 10 11 for the spatial model, which contributed much more information than R and K factors did. See Page 14 Line 3-7). Second, we introduced a new regional soil 12 erosion assessment method combining sample survey and geostatistics, which is 13 valuable for regions with limited input data or limited data resolution. In the 14 introduction part, we reviewed four methodologies for assessing regional soil 15 erosion including a) fractional scoring, b) plot measurements, c) field-based 16 17 approach, and d) model-based approach. Three kinds of (R)USLE-based approach include 1) area sample survey approach used by NRI in the USA, 2) raster 18 multiplication used by Europe, Australia, and many other regions and 3) sample 19 survey and geostatistics approach used in the fourth census on soil erosion in China, 20 which was introduced in this study. All these methodologies have their suitability 21 and limitations as discussed in the introduction part (Page 3 Line 21-28; Page 4, all 22 lines; Page 5, Line 1-15). Raster multiplication is a popular model-based approach 23 due to its lower cost, simpler procedures and easier explanation of result map. If the 24 resolution of input data for the entire region is enough to derive all the erosion 25 factors, raster multiplication approach is the best choice. However, there are several 26 concerns about raster multiplication approach: (1) The information for the support 27 practices factor (P) in the USLE is not easy to be collected given the common 28 29 image resolution and was not included in some assessments (Lu et al., 2001; Rao et 30 al., 2015), in which the resulting maps don't reflect the condition of soil loss but the risk of soil loss. Also, without the information of P factor, it is impossible to assess 31 the benefit from the soil and water conservation practices. (2) The accuracy of soil 32 erosion estimation for each cell is of concern if the resolution of database used to 33 derive the erosion factors is limited. Thomas et al. (2015) showed that the range of 34 LS factor values derived from four sources of DEM (20 m DEM generated from 35 1:50,000 topographic maps, 30 m DEM from ASTER, 90 m DEM from shuttle 36 radar topography mapping mission (SRTM) and 250 m DEM from global multi-37 resolution terrain elevation data (GMTED)) were considerably different. As we 38 mentioned in the supplement of SC2: 'Response to main comments by Referee #2' 39 (Shuiqing Yin, 17 Oct 2016), a recent research by our group showed that the slope 40 41 steepness based on the 30 m ASTER GDEM V1 is about 64% lower and the slope 42 length on the other hand was 265% larger, compared with the reference value based on the topography map with a scale of 1:2000 for a mountainous watershed in 43 Northern China. For many regions in the world, data used to derive erosion factor 44

1 such as conservation practice factor is often not available for all area, or the

- 2 resolution is not adequate for the assessment, as the referee has mentioned.
- 3 Therefore, the assessment method combining sample survey and geostatistics
- 4 proposed in this study is valuable. We added the information above in the revision
- 5 to make it more clear (Page 4 Line 11-21 ).
- 6
- 7

**8 AUTHOR RESPONSE TO RC #2**

**9 Main comments:**

1. The authors present in the introduction a number of methodologies for assessing 10 soil erosion: a) factorial scoring b) plot measurements c) field-based approach d) 11 Modelling (RUSLE). Then they analyse more in detail the application of RUSLE 12 as 3 different options: 1) sample survey 2) raster multiplication 3) sample survey 13 and geostatistics. The authors have followed the third option. Find below the most 14 15 important remarks and issues that authors should address in their revision: First remark: I would appreciate if the authors have compared their results with the 16 second option. This would give much more advanced knowledge in the 17 manuscript. You mentioned that you have available K-factor, R-factor maps at 18 250m resolution plus a land use map at 100000 scale. So, it would have been 19 excellent to compare your results with an estimated Soil loss by water erosion 20 (simply multiplying the above mentioned high resolution grids). 21 **Response:** It is not difficult to conduct raster layer multiplication technically, 22 however, we think the multiplication of R and K factors (assuming L=1, S=1, 23 B=1, E=1, T=1) reflects the potential of soil erosion, which is different from the 24 25 soil erosion estimated in this study. Therefore, no revision was made here. 26 27 2. As the 1st reviewer said (and I agree), the authors have presented an interpolation 28 method which takes into account 5 different group of parameters. It is logical (and obvious that the IV and V would perform much better than the I. In a recent 29 research (to be online soon), we identified cover management factor as the most 30

sensitive for estimating soil loss by water erosion. The manuscript could be even
more worthy if the authors have compared their findings with alternative methods
(plot measurements, expert knowledge, field-based approach).

- Response: In the original manuscript, we did some comparison with the plot
   measurements (Guo et al., 2015). We added more comparison with the result of
   the first three nationwide soil erosion investigations over China (Page 15 Line 19-
- 37 25) and the observation on the soil erosion rate for the forest and sediment
- 38 discharge for two watersheds (Page 16 Line 20-22).
- 39 40
- 41 3. The findings regarding the forests are much too high. Erosion of > 3 t ha-1 in
- 42 forest is not at all acceptable. Even there can be very steep slopes, the forestland

experience erosion of much less than 1 t ha-1 annually. Their comparison with the 1 findings of Guo (2015) and the findings in Europe (2015) show that erosion in 2 3 forests is much less. The same applies for grasslands. Please consider also a comparison of your findings with the paper of Wang et al (2016) "Assessment of 4 soil erosion change and its relationships with land use/cover change in China from 5 6 the end of the 1980s to 2010". 7 **Response:** Wang et al. (2016) used a factorial scoring method to assess soil erosion risk and change in China from the end of the 1980s to 2010. As it was 8 discussed in the introduction of this study, the resulting map by the factorial 9 scoring method depicts classes ranging from very low to very high erosion or 10 erosion risk. However, it can't generate soil erosion rates. We are also concerned 11 about the relatively high erosion rates of forest and grassland in the Shaanxi 12 13 province compared with some previous research. Our analyse showed that they came from the Primary Sampling Units, and was not introduced by the spatial 14 interpolation process. Possible reasons include: the different definitions of forest 15 and grassland, concentrated storms with intense rainfall, the unique topography in 16 17 Loess plateau and the sparse vegetation cover due to intensive human activities (Zheng and Wang, 2014). The minimum canopy density (crown cover) threshold 18 for the forest across the world vary from 10-30% (Lambrechts et al., 2009) and a 19 threshold of 10% was used in this study, which suggests on average a lower cover 20 coverage and higher B factor. Annual average precipitation varies between 328-21 1280 mm in Shaanxi, with 64% concentrating in June through September. Most 22 rainfall comes from heavy storms of short duration, which suggests the erosivity 23 density (rainfall erosivity per unit rainfall amount) is high. The slope degree and 24 slope length for the forest and grassland in Shaanxi province have been discussed 25 in the original manuscript (Page 16 Line 18-20; 24-26). The grassland includes 26 the native and artificial grassland, with more intensive livestock and human 27 activities. The result from the observation for two watersheds showed that the 28 29 erosion rate in the forest estimated in this study was consistent with the 30 observation. More discussion has been added in the revision (Page 16 Line 8-26). 31

32

4. Authors should explain and justify the selection of their statistical model BPST 33 34 and not the selection of Cubist or GPR or regression krigining? Moreover, In your geo-statistical model, the topography is ignored. Why? 35 **Response:** In spatial data analysis, there are mainly two approaches to make the 36 prediction of a target variable. One approach (e.g., kriging) treats the value of a 37 target variable at each location as a random variable and uses the covariance 38 function between these random variables or a variogram to represent the 39 correlation; another approach (e.g., spline or wavelet smoothing) uses a 40 41 deterministic smooth surface function to describe the variations and connections 42 among values at different locations. Our work (Bivariate Penalized Spline over Triangulation, BPST) takes the second approach. The relationship between the 43 traditional spatial statistics, and splines have been discussed in the literature, e.g. 44

Matheron (1981) and Wahba (1990). A brief comment is presented in the 1 following. Specifically, as discussed in Mitas and Mitasova (1999), "Kriging 2 assumes that the spatial distribution of a geographical phenomenon can be 3 modeled by a realization of a random function and uses statistical techniques to 4 analyze the data and statistical criteria for predictions. However, subjective 5 6 decisions are necessary such as judgement about stationarity, choice of function for theoretical variogram, etc. In addition, often the data simply lack information 7 about important features of the modelled phenomenon, such as surface analytical 8 properties or physically acceptable local geometries." In contrast, "Splines rely 9 on a physical model with flexibility provided by change of elastic properties of 10 11 the interpolation function. Often, physical phenomena result from processes which minimize energy, with a typical example of terrain with its balance 12 between gravitation force, soil cohesion, and impact of climate. For these cases, 13 splines have proven to be rather successful." For our problem, we also pay special 14 attention to the following two practical issues: (1) the data are not necessarily 15 evenly distributed; observations can be dense at some locations while sparse at 16 17 others; (2) the domain for the data can take non-rectangular shapes. In this work we introduce bivariate splines on triangulations to handle irregular 18 domains and propose to extend the idea of univariate penalized splines (Eilers 19 and Marx, 1996) to the two-dimensional case. The BPST method we consider 20 have several advantages. First, it provides good approximations of smooth 21 functions over complicated domains. Second, the computational cost for spline 22 evaluation and parameter estimation are manageable. Third, the BPST doesn't 23 require the data to be evenly distributed or on regular-spaced grid. 24 Topography factors based on smaller scale of topography map in the mountainous 25 and hilly area have large uncertainties. A recent research by our group showed 26 27 that the slope steepness based on the 30 m ASTER GDEM V1 is about 64% lower and the slope length on the other hand was increased by 265%, compared 28 29 with the reference value based on the topography map with a scale of 1:2000 for a 30 mountainous watershed in Northern China. We haven't obtained topography map with such high resolution yet. If larger scale topography map could be collected, 31 it is not difficult to incorporate topography factors into our model by adding L 32 and S factors in the equations (8) and (9). We added the information above in the 33 34 revision to make it more clear (Page 9 Line 3-9). 35

5. The field survey (section 2.1) indicates that the sampling of erosion points was 36 not so dense. Please give some levels of uncertainty taking into account that you 37 sampled on PSU every 25 km2 even less. Moreover, you mentioned that "PSU 38 points were surveyed", you don't describe how you estimate the R, K, LS, B, E, T 39 factors in each point? Did you sample and analyze the soil for estimating K-40 41 factor? Did you install a high temporal resolution rainfall station for measuring R-42 factor? Etc. Maybe this is somehow written in section 2.3 but it is not clear as you don't provide detailed information on how the R-factor, K-factor was calculated. 43 In the same way that you criticize the non-availability of all input layers when 44

multiplying the grids (factors), somebody can criticize your methodology that non 1 all information (K-factor, R-factor, ect) is available at point level. How you 2 3 respond to this? **Response:** The density of sample units in our survey depends on the level of 4 uncertainty and the budget of the survey. We tested sample density of 4% in four 5 6 experimental counties in different regions over China and found a density of 1% 7 was acceptable given the current financial condition (Page 7 Line 4-7). Lai and Wang (2013) provided the asymptotic properties of the BPST method, For 8 example, they investigated how the bias and variance of the BPST estimator 9 change with respect to the sample size and the number of the triangulations. Since 10 11 our data are a little sparse in some area, we employed the roughness penalties to regularize the spline fit; see the energy functional defined in equation (12). When 12 the sampling is sparse in certain area, the direct BPST method may not be 13 effective since the results may have high variability due to the small sample size. 14 The penalized BPST is more suitable for this type of data because it can help to 15 regularize the fit (Page 12 Line 7-9). 16 17 We added more information about how we estimated the R, K, LS, B, E, and T factors in each point (PSU) (See Page 9 Line 17-25). We didn't install a rainfall 18 station or collect soil samples for measuring R or K factor for each PSU. Instead, 19 we collected 2678 weather and hydrologic stations with erosive daily rainfall 20 from 1981 through 2010 and generated the R factor raster map over the entire 21 China (Xie et al., 2016). And for the K factor, the physicochemical data of 16,493 22 soil samples (which belong to 7764 soil series, 3366 soil families, 1597 soil 23 subgroups and 670 soil groups according to Chinese Soil Taxonomy) from the 24 Second National Soil Survey in 1980s and the latest soil physicochemical data of 25 1065 samples through the ways of field sampling, data sharing and consulting 26 27 literatures were collected to generate the K factor for the entire country (Liang et al., 2013; Liu et al., 2013). The R and K factors for each PSU were clipped 28 29 from the map of the entire country. A topography contour map with a scale of 30 1:10000 for each PSU was collected to derive the slope length and slope degree and to calculate the slope length factor and slope steepness factor (Fu et al., 31 2013). The calculation of B, E and T was based on the field survey of each PSU. 32 As we know that R factor in USLE requires the breakpoint rainfall data more than 33 34 20 years, which is not feasible if a rainfall station was set up in each PSU. We assumed that the variation of R factor could be captured by more than 2000 35 stations over China, which were the most stations we could collect at present. Soil 36 maps with scales of 1:500,000 to 1:200,000 (for different provinces) generated 37 more than 0.18 million polygons of soil attributes over mainland China, which 38 was the best available spatial resolution of soil information we could collect at 39 present. We assumed the result of the soil survey could be used to estimate the K 40 41 factor in our soil erosion survey. R factor and K factor in the sample point of NRI 42 USA was also from the interpolation result of weather stations and soil survey 43 map, respectively. We added the information above in the revision (See Page 9 Line 17-25). 44

**1 More specific comments related to text: 2 - P2L25-26: Rephrase the sentence. 3 **Response:** Followed the suggestion and revised it (Page 2 Line 26-27). 4 5 - P2L3: The reference should be Panagos et al, 2016a. The paper of Panagos et al, 6 7 2015a in your reference list does not feature in the literature. Please check carefully your literature. The same in P2L6 (it should be Panagos et al 2015; Panagos et al., 8 2016a). 9 10 **Response:** We supposed the referee refers P3L3 and P3L6. They were corrected and one reference missing in the old version was added. (Evans, R., and Boardman, J.: 11 12 The new assessment of soil loss by water erosion in Europe. Panagos P. et al., 2015 Environ. Sci. Policy 54, 438–447—A response. Environ. Sci. Policy, 58, 11-15, 13 2016.) They were revised (Page 3 Line 4-7). 14 15 16 - P3L16-17: You cannot put in the same importance the papers of Bosco (2015) and Panagos (2015) regarding the European soil erosion assessments. The second one is 17 much more advanced with new knowledge. For more info about the model evolution 18 in Europe, please consider the paper of Borrelli et al (2016), Land Use policy. 19 **Response:** Thanks for your information. We reviewed the paper of Borrelli et al. 20 (2016), Effect of good agricultural and environmental conditions on erosion and soil 21 organic carbon balance: A national case study, land use policy, 2016(50): 408-421) 22 and the information provided in the paper has been added in the revision (Page 3 Line 23 16-20). 24 25 26 - P3L23-25: some references to NRI methodology and the outcome results are missing here. I would also appreciate some applications and datasets derived (With 27 28 references) from this methodology. 29 **Response:** The 2012 NRI is the current NRI data, which provides nationally consistent data on the status, condition, and trends of land, soil, water, and related 30 resources on the Nation's non-Federal lands for the 30-year period 1982-2012. 31 USDA-NRCS (2015) summarized the results from the 2012 NRI, which also include 32 33 a description of the NRI methodology and use. A summary of NRI results on rangeland is presented in Herrick et al. (2010). See for example Brejda et al. (2001), 34 Hernandez, M., et al. (2013) for some applications using NRI data. See the revision in 35 Page 3 Line 23-30; Page 4 Line 1-4. 36 37 - P4L6-9: The European assessments was commented by 2 papers (you have put the 3 38 comments here) but you ignore the response of Panagos et al (2016a, 2016b) to those 39 3 critiques. 40 **Response:** Panagos et al. (2006a, 2016b) argued that field survey proposed by Evans 41 et al. (2015) is not suitable for the application at the European scale mainly due to 42**

work force and time requirements. They emphasized that the focus of their work is on 43

1 the differences and similarities between regions and countries across the Europe and

2 RUSLE model with the simple transparent structure can achieve their goal if

- 3 harmonized datasets were inputted. We added this in the revision. The revision is in
- 4 Page 4 Line 27-30.
- 5

```
Section 2.2 and elsewhere: R-factor, K-factor and land use map. Please give some
citations and source of this dada.
```

8 **Response:** Thanks for your suggestion. We introduced the source of the data and

- 9 gave the Citations. The data for the R-factor was based on 2678 weather and
- 10 hydrologic stations with erosive daily rainfall from 1981 through 2010. The daily
- 11 model used to generate at-site R factor was Model I in **Xie et al. (2016**). The data of
- 12 K-factor was based on the physicochemical data of 16,493 soil samples from the

13 Second National Soil Survey in 1980s and latest soil physicochemical data of 1,065

14 samples through the ways of field sampling, data sharing and consulting literatures

15 (Liang et al., 2013; Liu et al., 2013). Land use map with a scale of 1:100000 was

- 16 from China's Land Use/cover Datasets (CLUD), which were updated regularly at a
- 17 five-year interval from the late 1980s through the year of 2010 with standard
- 18 procedures based on Landsat TM/ETM images (Liu et al., 2014). Land use map used
- in this study was the version of 2010. The revision is in Page 8 Line 17-28.
- 20

You may be familiar with CSLE but for some non-Chinese, it would be better to
write 2 lines about the biological and engineering factor.

**Response:** Good suggestion! We revise it (Page 8 Line 7-16). Biological (B),

Engineering (E) and Tillage (T) factor was defined as the ratio of soil loss from the

actual plot with biological, engineering or tillage practices to the unit plot. Biological

- 26 practices are the measures to increase the vegetation coverage for reducing runoff and
- soil loss such as trees, shrubs and grass plantation and natural rehabilitation of
- vegetation. Engineering practices refer to the changes of topography by engineering
- 29 construction on both arable and non-arable land using non-normal farming equipment
- 30 (such as earth mover) for reducing runoff and soil loss such as terrace, check dam and
- so on. Tillage practices are the measures taken on the arable land during ploughing,
- harrowing and cultivation processes using normal farming operations for reducing
- runoff and soil loss such as crop rotation, strip cropping and so on (Liu et al., 2002).
- 34

- P7L11-13: Your method has many similarities with the estimation of C-factor in
Europe based on vegetation density and land use (Panagos et al., 2015 Land use

- 37 policy).
- **Response:** Yes. Both C-factor in Panagos et al. (2015) and B-factor in this study for forest, shrub land and grassland were estimated based on the vegetation density
- 40 derived from satellite images. The difference is that C factor in Panagos et al. (2015)

41 for arable land and non-arable land was estimated separately based on different

- 42 methodologies, whereas in this study, the biological factor (B factor) was used to
- 43 reflect biological practices on the forest, shrub land or grassland for reducing runoff
- and soil loss and the tillage factor (T factor) was used to reflect tillage practices on the

farmland for reducing runoff and soil loss. For the farmland, biological factor equals 1
and for the other land uses, tillage factor equals 1. Revision is in Page 9 Line 19-27.

- 3
- P7 Equations 2 and 3: they seem to be very similar. Which is the difference, please
  explain.

6 **Response:** Yes, they were similar because both of them used weight-averaged method

7 by the area of plots. The difference is Equation 2 is for the estimation of soil loss for

- 8 each land use in the PSU and Equation 3 is for the estimation of soil loss for the entire
- 9 PSU. Revision is in Page 10 Line 1-7.
- 10

- Model I has no sense as it is obvious to be so poor!!!

- Response: Model I was a naive method which was used as a comparison (Page 10Line 10-11).
- 13 14

- Section 2.4.2: you start without any introduction about BPST model. Please add anintroductory paragraph

17 Response: Good suggestion. We will add an introduction paragraph about Bivariate Penalized Spline over Triangulation (BPST) method as follows in the revision: "In 18 spatial data analysis, there are mainly two approaches to make the prediction of a 19 target variable. One approach (e.g., kriging) treats the value of a target variable at 20 each location as a random variable and uses the covariance function between these 21 random variables or a variogram to represent the correlation; another approach (e.g., 22 23 spline or wavelet smoothing) uses a deterministic smooth surface function to describe the variations and connections among values at different locations. In this study, 24 Bivariate Penalized Spline over Triangulation (BPST), which belongs to the second 25 approach, was used to explore the relationship between location information in a two-26 dimensional (2-D) domain and the response variable. Given the complexity of the 27 boundary in our data, traditional methods of smoothing which rely on the Euclidean 28 29 metric or which measures smoothness over the entire real plane may then be 30 inappropriate; see excellent discussions in Ramsay (2002) and Wood et al (2008). Here we present a method using bivariate splines over triangulations to smooth scattered 31 bivariate data over domains with complex boundaries. The smoothing function is the 32 minimum of a penalized sum-of-squares error functional. To be more specific, Let 33  $(x_i, y_i) \in \Omega$  be the latitude and longitude of unit i for i = 1, 2, ..., n... "Revision is in 34 Page 11 Line 20-24; Page 12 Line 1-9. 35

36

P11L13-15: it is obvious that model I and II will have no extreme erosion levels (as
the land use is ignored). The smoothening effect is obvious!

**Response:** It is true that when the land use is ignored, the extreme erosion levels,

40 mostly in farmland and bare land, were smoothed by the surrounding low erosion

41 levels, mostly in forest, shrub land, grassland and construction land (Page 14 Line 16-

- 42 18).
- 43

- Conclusions: P14L5-6: it is better somewhere in the introduction. In general the

conclusions should highlight the important findings of this study. 1 **Response:** Good suggestion. We revised it (Page 6, Line 9-18; Page 17, Line 18-25). 2 Many studies indicated that land use is one of the most important factors in soil 3 erosion estimation. Our study quantified how important the knowledge of land use, 4 rainfall erosivity and soil erodibility are for estimating regional soil loss by comparing 5 6 five different spatial models. Besides, we introduced a new model-based regional soil 7 erosion assessment method, which is valuable when input data used to derive soil erosion factors is not available for the entire region, or the resolution is not adequate. 8 9 - The way forward: The authors should conclude about the usefulness of their 10 methodology. How this can be used? How it can be complementary to the traditional 11 multiplication of grids. 12 13 Response: Good suggestion! We have add more information in the revision (Page 18 Line 4-16. When the input data used to derive soil erosion factors is not available or 14 the resolution is not adequate, and the budget is limited, our suggestion is sampling a 15 certain amount of small watersheds as primary sampling units and put the limited 16 17 money into these sampling units to ensure the accuracy of soil erosion estimation in these sampling units. Limited money could be used to collect high resolution of data 18 such as satellite images and topography maps and conduct field survey to collect 19 information such as conservation practices for these small watersheds. Then use the 20 best available raster layers for the entire region and construct the spatial model by 21 aggregating them as a factor Q. 22 23 Fig1: The Dark green image Towner-ship cannot be 10 x 40km? 24 **Response:** It was a typo and it has been revised (Fig. 2). Thank you! 25 26 27 Fig2: The land uses are 5 and not 3. Response: The categories of land use are three including the farmland, forest and 28 29 residential area. There are five plots, two of which are farmland, two are forest and 30 one is residential area. More explanation was added in the revision (Fig 3). 31 Fig 5: It is difficult to see the distinction between 40-80 and 80 categories in the first 32 bar. 33 **Response:** The percentage of over 80 t ha-1y-1 is small. A different color has been 34 used in the revision (Fig. 7). 35 36 Fig. 7: Scale bar and overview map is missing. As a non-Chinese, I don't know where 37 this province is located. 38 **Response:** A figure with an overview map has been added in the revision (Fig.1 has 39 been added). Thank you! 40 41

42 Minor comments:

| 1  | - P5  | L7: please replace with "erodibility"                                                               |
|----|-------|-----------------------------------------------------------------------------------------------------|
| 2  | Res   | ponse: There is a typo on P5L7 (erobility should be erodibility). Revised. Thanks!           |
| 3  |       |                                                                                                     |
| 4  | - ''s | oil species" is not a mature term                                                                   |
| 5  | Res   | ponse: We changed "soil species" into "soil series" according to Chinese Soil                |
| 6  | Tax   | onomy, in which the classification system is Order-Suborder-Group-Subgroup-                         |
| 7  | Fan   | nilies-Series (Gong and Zhang, 2007). Page 8 Line 26.                                               |
| 8  |       |                                                                                                     |
| 9  | - 3.  | 1: Four or Five models?                                                                             |
| 10 | Res   | ponse: It was a typo and it was five models. It has been corrected in the revision.          |
| 11 |       |                                                                                                     |
| 12 | Ref   | erences:                                                                                            |
| 13 | [1]   | Borrelli P., Paustian K., Panagos P., Jones A., Schütt, B., Lugato, E.: Effect of good agricultural |
| 14 |       | and environmental conditions on erosion and soil organic carbon balance: A national case study.     |
| 15 |       | Land use policy, 50: 408-421, 2016.                                                                 |
| 16 | [2]   | Brejda, John J., et al. "Estimating surface soil organic carbon content at a regional scale using   |
| 17 |       | the National Resource Inventory." Soil Science Society of America Journal 65.3 (2001): 842-         |
| 18 |       | 849.                                                                                                |
| 19 | [3]   | Eilers, P. H., & Marx, B. D. (1996). Flexible smoothing with B-splines and penalties. Statistical   |
| 20 |       | science, 89-102.                                                                                    |
| 21 | [4]   | Evans, R., Collins, A. L., Foster, I. D. L., Rickson, R. J., Anthony, S. G., Brewer, T., Deeks, L., |
| 22 |       | Newell-Price, J. P., Truckell, I. G., and Zhang, Y.: Extent, frequency and rate of water erosion    |
| 23 |       | of arable land in Britain-benefits and challenges for modelling. Soil Use Manage, in press,         |
| 24 |       | 2015.                                                                                               |
| 25 | [5]   | Fu, S. H., Wu, Z. P., Liu, B. Y., and Cao, L. X.: Comparison of the effects of the different        |
| 26 |       | methods for computing the slope length factor at a watershed scale. Int Soil Water Conserv          |
| 27 |       | Res, 1(2), 64-71, 2013.                                                                             |
| 28 | [6]   | Guo, Q. K., Hao, Y. F., and Liu, B. Y.: Rates of soil erosion in China: A study based on runoff     |
| 29 |       | plot data. Catena, 24, 68-76, 2015.                                                                 |
| 30 | [7]   | Gong, Z. T., and Zhang G. L.: 2007. Chinese soil taxonomy: A milestone of soil classification       |
| 31 |       | in China. Science Foundation in China, 15(1): 41-45.                                                |
| 32 | [8]   | Hernandez, M., et al. "Application of a rangeland soil erosion model using National Resources       |
| 33 |       | Inventory data in southeastern Arizona." Journal of Soil and Water Conservation 68.6 (2013):        |
| 34 |       | 512-525.                                                                                            |
| 35 | [9]   | Herrick, Jeffrey E., et al. "National ecosystem assessments supported by scientific and local       |
| 36 |       | knowledge." Frontiers in Ecology and the Environment 8.8 (2010): 403-408.                           |
| 37 | [10]  | Lai, M. J., and Wang, L.: Bivariate penalized splines for regression. Statistica Sinica, 23, 1399-  |
| 38 |       | 1417, 2013.                                                                                         |
| 39 | [11]  | Lambrechts, C., Wilkie, M. L., and Rucevska, I.: Vital forest graphics, UNEP/GRID-Arendal,          |
| 40 |       | 2009.                                                                                               |
| 41 | [12]  | Liang, Y., Liu, X. C., Cao, L. X., Zheng, F. L., Zhang, P. C., Shi, M. C., Cao, Q. Y., and Yuan,    |
| 42 |       | J. Q.: K value calculation of soil erodibility of China water erosion areas and its Macro-          |
| 43 |       | distribution. Soil and Water Conservation in China, 10, 35-40, 2013 (in Chinese with English        |

[revised manuscript text omitted]

- 2 Panagos, P., Borrelli, P., Poesen, J., Meusburger, K., Ballabio, C., Lugato, E., Montanarella, L., and Alewell, C.:
- Reply to "The new assessment of soil loss by water erosion in Europe. Panagos P. et al., 2015 Environ. Sci.
  Policy 54, 438–447—A response" by Evans and Boardman [Environ. Sci. Policy 58, 11–15]. Environ Sci
  Policy, 59, 53–57, 2016a.
- Panagos, P., Borrelli, P., Poesen, J., Meusburger, K., Ballabio, C., Lugato, E., Montanarella, L., and Alewell, C.:
  Reply to the comment on "The new assessment of soil loss by water erosion in Europe" by Fiener & Auerswald. Environ Sci Policy, 57, 143–150, 2016b.
- Rao, E. M., Xiao, Y., Ouyang, Z. Y., and Yu, X. X.: National assessment of soil erosion and its spatial patterns in
   China. Ecosystem Health and Sustainability, 1(4), 13, doi: 10.1890/EHS14-0011.1, 2015.
- Renard, K. G., Foster, G. R., Weesies, G. A., McCool, D. K., and Yoder, D. C.: Predicting soil erosion by water.
   U.S. Department of Agriculture, Agricultural Research Service, Agriculture Handbook 703, Washington DC, 1997.
- Renschler, C. S., and Harbor, J.: Soil erosion assessment tools from point to regional scales—the role of
   geomorphologists in land management research and implementation. Geomorphology, 47, 189–209, 2002.
- Singh, G., Babu, R., Narain, P., Bhushan, L. S., and Abrol, I. P.: Soil erosion rates in India. J Soil Water Cons, 47
  (1), 97–99, 1992.
- Thomas, J., Prasannakumar, V., and Vineetha, P.: Suitability of spaceborne digital elevation models of different
   scales in topographic analysis: an example from Kerala, India. Environ Earth Sci, 73, 1245-1263, 2015.
- 20 USDA: Summary report: 2012 National Resources Inventory. National Resources Conservation Service,
   21 Washington DC, and Center for Survey Statistics and Methodology, Iowa State University, Ames, Iowa,
   22 2015.
- 23
- Van der Knijff, J. M., Jones, R. J. A., and Montanarella, L.: Soil erosion risk assessment in Europe. European
   Commission, EUR 19044 EN, Luxembourg, 2000.
- 26 Vrieling, A.: Satellite remote sensing for water erosion assessment: A review. Catena, 65, 2–18, 2006.

Wang, G., and Fan, Z.: Study of Changes in Runoff and Sediment Load in the Yellow River (II). Yellow River
 Water Conservancy Press, Zhengzhou, China, 2002 (in Chinese).

Wang, X., Zhao, X. L., Zhang, Z. X., Li, L., Zuo, L. J., Wen, Q. K., Liu, F., Xu, J. Y., Hu, S. G., and Liu, B.:
 Assessment of soil erosion change and its relationships with land use/cover change in China from the end of
 the 1980s to 2010. Catena, 137, 256-268, 2016.

- Wischmeier, W. H., and Smith, D. D.: Predicting Rainfall Erosion Losses: A Guide to Conservation Planning.
   U.S. Department of Agriculture, Agricultural Research Service, Agriculture Handbook 537, Washington DC,
   1978.
- Wischmeier, W. H., and Smith, D. D.: Predicting rainfall-erosion losses from cropland east of the Rocky
   Mountains. U. S. Department of Agriculture, Agricultural Research Service, Agriculture Handbook 282,
   Washington DC, 1965.

- Xi, Z. D., Sun, H., and Li, X. L.: Characteristics of soil erosion and its space-time distributive pattern in southern
   mountains of Shaanxi province, Bull Soil Water Conserv, 17(2), 1–6, 1997 (in Chinese with English abstract).
- Xie, Y., Yin, S. Q., Liu, B. Y., Nearing M., and Zhao, Y.: Models for estimating daily rainfall erosivity in China.
   J Hydrol, 535, 547–558, 2016.
- 5 Zheng, F. L., and Wang, B.: Soil Erosion in the Loess Plateau Region of China. Tsunekawa A. et al. (eds.),
- Restoration and Development of the Degraded Loess Plateau, China, Ecological Research Monographs.
  Springer, Japan, doi.10.1007/978-4-431-54481-4\_6, 2014.
- 8

**1 Tables**

|       | Land use and sample size  |      |            |                               |       |           |         |  |
|-------|---------------------------|------|------------|-------------------------------|-------|-----------|---------|--|
| Model | Farmland Forest Shrub lar |      | Shrub land | d Grassland Construction land |       | Bare land | Overall |  |
|       | 1134                      | 1288 | 573        | 683                           | 401   | 32        | 4111    |  |
| Ι     | —                         | _    | —          | _                             | —     | —         | 352.5   |  |
| II    | —                         | _    | —          | —                             | —     | —         | 345.5   |  |
| III   | 399.7                     | 25.3 | 45.5       | 20.0                          | 165.7 | 4264.6    | 177.2   |  |
| IV    | 404.3                     | 25.3 | 45.4       | 19.5                          | 164.5 | 3691.2    | 173.8   |  |
| V     | 365.4                     | 24.3 | 38.0       | 16.3                          | 162.5 | 3555.1    | 152.9   |  |

2 Table 1. Mean squared errors of soil loss (A) using bivariate penalized spline over triangulation (BPST)

1 Table 2. Soil loss rates (t ha-1y-1) for the farmland, forest, shrub land and grassland by Model V in this study and in

|                   | Land use                 | Mean  | Standard deviation |
|-------------------|--------------------------|-------|--------------------|
| This study        | Farmland                 | 19.00 | 17.94              |
|                   | Forest                   | 3.50  | 2.78               |
|                   | Shrub land               | 10.00 | 7.51               |
|                   | Grassland                | 7.20  | 5.23               |
| Guo et al. (2015) | Farmland (Conventional)  | 49.38 | 57.61              |
|                   | Farmland (Ridge tillage) | 19.27 | 13.35              |
|                   | Farmland (Terracing)     | 0.12  | 0.28               |
|                   | Forest                   | 0.10  | 0.12               |
|                   | Shrub land               | 8.06  | 7.47               |
|                   | Grassland                | 11.57 | 12.72              |

2 Northwest region of China from Guo et al. (2015).

| Ductostuno |                           | Amount $(10^6 \text{ t y}^{-1})$ | Rate
(t ha -1 y -1 ) | Source (%) |        |               |               |
|------------|---------------------------|----------------------------------|-----------------------------------------------|------------|--------|---------------|---------------|
| city       | Area (10 4 ha) |                                  |                                               | Farmland   | Forest | Shrub
land | Grass
land |
| Xi'an      | 100.4                     | 6.3                              | 6.3                                           | 52.9       | 11.6   | 7.9           | 20.6          |
| Ankang     | 230.0                     | 26.6                             | 11.6                                          | 42.8       | 10.7   | 2.8           | 42.7          |
| Baoji      | 178.5                     | 13.2                             | 7.4                                           | 39.3       | 15.1   | 7.5           | 37.9          |
| Hanzhong   | 266.7                     | 21.8                             | 8.2                                           | 42.5       | 12.3   | 3.6           | 40.2          |
| Shangluo   | 193.0                     | 8.5                              | 4.4                                           | 68.0       | 13.1   | 5.9           | 12.9          |
| Tongchuan  | 38.6                      | 3.7                              | 9.6                                           | 37.9       | 7.8    | 23.6          | 28.5          |
| Weinan     | 129.5                     | 6.4                              | 5.0                                           | 54.4       | 3.9    | 9.5           | 26.7          |
| Xianyang   | 101.0                     | 5.2                              | 5.2                                           | 44.4       | 8.2    | 8.9           | 35.3          |
| Yan'an     | 364.9                     | 55.9                             | 15.3                                          | 54.5       | 3.1    | 12.1          | 30.0          |
| Yulin      | 427.7                     | 50.9                             | 11.9                                          | 51.4       | 2.6    | 3.7           | 40.4          |
| Overall    | 2030.4                    | 198.7                            | 9.8                                           | 49.8       | 6.8    | 7.1           | 35.0          |

1 Table 3. Annual soil loss amount, rate and main sources by Model V for ten prefecture cities in Shaanxi province.

| 1                                                                                                       | Table 4 Soil erosion rate for the forest and sediment discharge for two watersheds |                            |                                     |                              |                                           |                          |                                           |  |
|---------------------------------------------------------------------------------------------------------|------------------------------------------------------------------------------------|----------------------------|-------------------------------------|------------------------------|-------------------------------------------|--------------------------|-------------------------------------------|--|
|                                                                                                         | Area                                                                               |                            | Runoff                       | Sediment
discharge | Soil loss
rate                         | Percent of forest | Soil loss rate                            |  |
|                                                                                                         |                                                                                    | (104 ha) | $(10^9 \text{ m}^3 \text{ y}^{-1})$ | $(10^{6} t y^{-1})$          | (t ha-1 y-1) | (%)               | (t ha-1 y-1) |  |
|                                                                                                         | Jinghe a                                                                | 454.2               | 1.837                        | 246.7                        | 54.3                               | 6.5               | 19.0                               |  |
|                                                                                                         | Luohe b                                                                 | 284.3               | 0.906                        | 82.6                  | 29.1                               | 38.4              | 1.3~2.1                            |  |
| 2                                                                                                       | a. Based or                                                                        | n the observat             | ion at Zhangjiash                   | an hydrologica               | 1 station from 1                          | 950 through 198          | 89.                                |  |
| 3 b. Based on the observation of at Zhuanghe hydrological station from 1959 through 1989. |                                                                                    |                            |                                     |                              |                                           |                          | 9.                                 |  |
| 4                                                                                                       |                                                                                    |                            |                                     |                              |                                           |                          |                                           |  |

**2 Figures**

---

## Referee Report (RR1)

I am surprised that authors get back after long time. Authors have taken into account my comments but there are still some issues to resolve. I would propose a major revision as I have seen the comments from the other reviewer.

Major issues:

- Authors did not convince both reviewers that it is better to use their proposal instead of applying an USLE with the available resolution of input data.
- The erosion rates in forest lands are very extreme and not justified. This can be connected to the erroneous outputs from NDVI calculations (I am very concerned about the NDVI. This is also confirmed by the difference with the study of Guo.
- The whole issue with LS factor and how authors present it. Previous studies in European LS-factor and the study of Borrelli in soil erosion in Italy demonstrated that the finer the resolution, the best are the LS-factor estimates and soil erosion model outputs. I do not agree with the what authors state in P12L13-14. The same confusion is created in P17L20-23. In the past studies in Europe it was shown that the finer the resolution, the best representation of the topography and then the more accurate the estimate of LS-factor and erosion rates.
- Table 1: quite confusion. What are the numbers in the row below the land uses? Moreover model numbers are missing in this table
- The low importance given in the model approach for R-factor is preoccupying.
- General comment about the methodology: Interpolating the soil erosion modelled point estimates is less appropriate than the application of model using input layers. Authors do not make clear statements of this and readers may be confused in making interpolation of erosion points even if they have the input layers. This point should be better addressed.

Some other issues:

- P2L15-15: The 2 sentences on water & wind erosion are repetitive
- P2L20. I don't agree with this sentence. It maybe that land use is improved and the soil erosion risk decreased.
- P3L3. Experts have always expertise
- P3L29: "seamless" is not appropriate.
- P5L3-5: The sentence is confusing
- P6L25L : for first time you present the factors R, K, L, S....B, E: and the reader does not know what those factors are. In the next page you describe them...but there should be a logical sequence.
- Eq (1): What is uk? Describe below..
- P7L27: soil families are not a good term
- Equation 2: What is q?
- What are the measurement units of factors in equation 2?
- L10P17: What is f(,)
- Tabl3e 5: What not provide the total soil loss and discuss on the comparison of soil loss with sediment loss?

---

## Editor Decision (ED1)

I agree with the authors and the reviewers, that the manuscript addresses an important topic: estimation of soil erosion rates in a large scale. As acknowledged by all, this task will lead to a number of methodological challenges.

The authors try to address those challenges by applying a spline-based interpolation technique, using different levels of additional external information, such as land use, rainfall, soil characteristics.

I feel that this can be a valuable approach in addressing the challenge of large scale erosion (maps). However, I think the authors do not show (enough) how superior their method is, compared to other techniques. Therefore, I suggest that the authors try to work along the remarks of the reviewers. I also agree with rev. #2 that one should compare with other independent soil erosion values, see also below.

I addition, I think that information about the values and the spatial distribution (maps) of the additional information (land use, rainfall, soil characteristics, topography) should be displayed. This needs to be given in the same spatial detail as the interpolation method uses this information.

I also wonder if the authors can give a measured/observed value of a lumped total erosion rate in a meso-scale region over some years (e.g. soil deposits in a reservoir) and compare such value(s) with their estimates. This would be particular helpful to get more confidence into the very high erosion rates in the forest.

Other comment: Please do not display internationally disputed borders, as you do in your Fig. 1a, page 12, in your "AC1: Final response to the comments from two referees". This is not acceptable for publication in a final version.

---

## Author Response (AR2)

We would like to thank the editor and two referees for their valuable comments. We revised the manuscript thoroughly taking into account all the comments from the editor and referees. Here is a detailed author response to all comments from the editor and referees. Please refer to the CLEAN VERSION in the supplement for the page and line information in the response.

**AUTHOR RESPONSE TO THE EDITOR**

1. Try to reach a more interesting balance of the paper, i.e. you can reduce the introduction according to ref #2 suggestions, and extend the discussion and conclusions regarding possibly (missing) validations and regarding sensitivity of input parameters etc.

**Response:** We have reduced the introduction into 3 pages and extended the discussion into 4 pages according to ref #2 suggestion. We have added the conclusions about the validation (MSE and Nash-Sutcliffe model efficiency coefficient, ME) and the sensitivity of topography factors derived from different resolutions of DEM data.

- Try to improve the validation of your approach. This is the essential core of your work. You may also consider to more clearly present (a) methods and (b) results and validation of results, see suggestions of ref. #2
   **Response:** We added ME to assess the performance of models together with the MSE (Page11 Line 22-25). We added a table (Table 2) to show the ME for all seven models per land use and a scatterplot (Figure 6) to show the deviation of the simulation from the observation for four main land uses. Method (Section 2.5), Result (Section 3.1) and Conclusion were revised accordingly.
- Please explain again, why you think that available slope data (30 m Aster or 90/30 m SRTM) are less relevant than large-distance interpolated R- and K- factors. And discuss the related uncertainty.

**Response:** We compared seven models based on Bivariate Penalized Spline over Triangulation (BPST) method to generate a regional soil erosion assessment from the PSUs in the revision. Among them, four models assisted by the land use and single erosion factor (Model II: land use and R; Model III: land use and K; Model IV: land use and L; Model V: land use and S) were compared with the model assisted by the land use only (Model I). Ten-fold cross-validation results based on the PSU data demonstrated that slope steepness factor derived from 1:10000 topography map is the best single covariate, reducing about 20% of the MSE for the interpolation of soil loss by comparing the model assisted by the land use and S factor with the model assisted by the land use. Soil erodibility and slope length information reduced about 10% of the MSE. Rainfall erosivity contribution is insignificant, with the MSE decreasing less than 1%.

In addition, since we don't have 1:10000 topography maps available for the entire region at present, we conducted a sensitivity analysis by preparing LS-factor from 30-m or 90-m SRTM DEM data and replacing the LS-factor derived from 1:10000

topography maps in the PSUs to detect if coarser resolution of topography data can be used as the covariate in the interpolation process. The result showed that the LS-factor derived from 30-m DEM or 90-m DEM deteriorated the estimation when they were used as the covariates together with the land use, R and K, with the MSEs increasing about two times than those for the model assisted by the land use! We think the finding is very interesting and important and added it in the manuscript (see Section 2.6 and Section 3.1 for details).

4. Please also follow comment #5 of ref. #2. This comment seems reasonable for me. Or - we both may be wrong - you may explain why the results of model I and II contains new knowledge, and why you think that a linear interpolation [excluding land-use boundaries] might be appropriate.

**Response:** Model I and Model II in the original manuscript were deleted and we redesigned seven models for the comparison (see Section 2.4.1 and Result for details).

**AUTHOR RESPONSE TO RC #1**

- 1. P3L4. It is Evans and not Evan **Response:** Revised.
- 2. P4L1: "2012 NRI is the current NRI data " ... this is not correct sentence..please rephrase.

**Response:** This sentence was deleted due to the simplification of the introduction.

3. - P4L15. Please add a reference in the sentence This is somehow importance and you should somehow use a literature reference for this

**Response:** This sentence was moved to the discussion part and a reference (Liu et al., 2013) was added.

- 4. P5L2: Panagos et al 2016a (and not 2006).**Response:** Revised.
- 5. P6L2: "It is important to note is...." It is not correct English **Response:** Revised.
- 6. In references, CORINE reference is not needed (neither in the text)...This is too old
   Response: Deleted.
- 7. In the text, you refer to the estimation of C-factor in Europe and you compare with yours but in the references you missed to add the reference of the European Cover management factor paper (in Land use policy).

**Response:** Reference to Panagos et al. (2015a) was added. Panagos, P., Borrelli, P., Meusburger, K., Alewell, C., Lugato, E., Montanarella, L.: Estimating the soil

erosion cover-management factor at the European scale. Land Use Policy, 48, 38–50, 2015a.

- 8. Table 1: It should be "Mean square error.....triangulation (BPST) per land use "
   Response: Revised.
- Table 3: Replace "rate" header with "Mean Rate" Response: Revised.
- 10. Figure 5. Attention in the units of K-factor and R-factor. A parenthesis is not well positioned.Response: Revised.

**AUTHOR RESPONSE TO RC #2**

- The structure of the paper is unbalance. The author provide an extensive introduction (approx. 5 pages; whereas it is not clear why all the different approaches to estimate regional/national erosion need to be presented here in so much detail), while the entire results and discussion is similar in length. A lot of information from the literature would be better placed/discussed in the discussion.
   **Response:** Firstly, we reduced the introduction to about three pages by deleting some information which was not closely related and moving some information to the discussion part. Secondly, we carried out a more in-depth discussion about the uncertainty of the proposed assessment method (Section 4.1) and the comparison with raster layer multiplication method (Page 17 Line 12-30). The discussion part is more than four pages after the revision.
- 2. The authors present an interesting interpolation scheme but the validation of their approach is weak (a few lines in chapter 3.1). In general, I would expect two major parts of the paper: (a) methods and (b) results and validation of results. I would, for example, expect different goodness-of-fit parameters as well as a more extensive discussion where the model performs appropriate and where major errors can be expected. Errors in interpolation and PSU data! The comparison with the data of Guo et al. (2015) (in discussion) shows that the results are in a similar rage but this is not a validation.

**Response:** We kept Mean squared prediction error (MSE) and added Nash-Sutcliffe model efficiency coefficient (ME) as the goodness-of-fit parameters to assess the performance of models. We added a table (Table 2) showing the ME for all seven models per land use and a scatterplot (Figure 6) showing the deviation of the simulation from the observation for four main land uses. Method (Section 2.5), Result (Section 3.1) and Conclusion were revised accordingly. A more extensive discussion (Section 4.1) on the uncertainty of the

assessment including possible errors in the PSU and interpolation were added.

- 3. Any kind of evaluation how sensitive the results are regarding quality of input data is missing? Which are the most important co-variables. At least some sensitivity analysis would be very helpful to underline the quality of the method. **Response:** We added a sensitivity analysis about topography factors derived from different resolutions of DEM data (1:10000 topography map with 5-m contour intervals, 30-m and 90-m SRTM DEM data) on the soil loss estimation since the topography factors are the dominant small scale modulators of soil erosion and the lack of the high resolution DEM data is often the case (Section 2.6 and Section 3.1). Seven models were designed (Section 2.4.1) and it showed that among the four erosion factors as the covariates, S factor derived from 1:10000 topography map contributed the most information, followed by K and L factors derived from 1:10000 topography map, and R factor made almost no contribution to the spatial estimation of soil loss. However, LS-factor derived from 30-m or 90-m SRTM DEM data worsened the estimation when they were used as the covariates for the interpolation of soil loss by increasing two times of the MSE. Due to the unavailability of 1:10000 topography map for the entire area in this study, the model assisted by the land use, R and K factor was used to generate the regional assessment of the soil erosion for Shaanxi province.
- 4. The authors argued that the available slope data (30 m Aster or 90/30 m SRTM) are not good enough to be included as co-variables in their interpolation. I agree that these data are far from perfect, but compared to an interpolated R factor, soil information derived from a relatively coarse map (K factor), I assume that the slope data show less uncertainty. As slope is one of the dominant small scale modulator of soil erosion (compared to all other data used) I disagree to omit slope as co-variable from the interpolation.

**Response:** We agree that slope is one of the dominant factors in the assessment of the regional soil loss, which was confirmed in this study. By comparing four models assisted by the land use and single erosion factor (Model II: land use and R; Model III: land use and K; Model IV: land use and L; Model V: land use and S) with the model assisted by the land use only (Model I), we quantified the relative importance of the erosion factors. The slope steepness factor derived from 1:10000 topography map is the best single covariate, reducing about 20% of the MSE for the interpolation of soil loss by comparing the model assisted by the land use and S factor with the model assisted by the land use. Soil erodibility and slope length information reduced about 10% of the MSE. Rainfall erosivity made almost no contribution with the MSE decreasing less than 1%. However, LS-factor derived from 30-m or 90-m SRTM DEM data worsened the estimation when they were used as the covariates for the interpolation of soil loss (see Section 2.4.1, Result and Conclusion for details).

5. I strongly suggest to remove model I and II from paper. This has two reasons: (a) It is obvious from the results (e.g. line 6-8 and line 17-19 on page 15 in tracked

changed document) that the interpolation without taking land use into account leads to an underestimation of erosion on farmland and an overestimation in forested areas. This is obvious and not worth to be published. Comparing models III to V with models I to II (e.g. Fig. 9) is misleading. (b) Land use produces discrete borders resulting in specific non-continuous changes in soil erosion. An interpolation without taking the 'steps' in erosion into account will always produce artificial results (and e.g. in geostatistics would violate general assumptions of the method).

**Response:** Model I and Model II in the original manuscript were deleted and we redesigned seven models for the comparison (Section 2.4.1), which were:

(1) Estimating A with the land use as the auxiliary information (Model I);

(2) Estimating A with R and land use as the auxiliary information (Model II);

(3) Estimating A with K and land use as the auxiliary information (Model III);

(4) Estimating A with L and land use as the auxiliary information (Model IV);

(5) Estimating A with S and land use as the auxiliary information (Model V);

(6) Estimating A with R, K and land use as the auxiliary information (Model VI);

(7) Estimating A with R, K, L, S and land use as the auxiliary information (Model VII).

MSE and ME from ten-fold cross-validation based on PSU data were used to compare and evaluate the performance of the models. Due to the unavailability of 1:10000 topography map for the entire area, 30-m DEM and 90-m DEM were also used to generate LS-factor and replace the LS-factor in Model VII to determine if it can be used as the covariate in the interpolation of soil loss (Section 2.6 and Section 3.1).

**Regional soil erosion assessment based on sample survey and geostatistics**

Shuiqing Yin1, 2, Zhengyuan Zhu3, Li Wang3, Baoyuan Liu1, 2, Yun Xie1, 2, Guannan Wang4
 and Yishan Li1, 2

1State Key Laboratory of Earth Surface Processes and Resource Ecology, Beijing Normal University, Beijing
 100875, China

7 2School of Geography, Beijing Normal University, Beijing 100875, China

8 3Department of Statistics, Iowa State University, Ames 50010, USA

9 4Department of Mathematics, College of William & Mary, Williamsburg 23185, USA

10 Correspondence to: Baoyuan Liu (baoyuan@bnu.edu.cn)

11

12

13 Abstract. Soil erosion is one of the majorost significant environmental problems in China. From 2010-2012 in-14 China, the fourth national census for soil erosion sampled 32,364 Primary Sampling Units (PSUs, small 15 watersheds) with the areas of 0.2-3 km2. Land use and soil erosion controlling factors including rainfall erosivity, 16 soil erodibility, slope length, slope steepness, biological practice, engineering practice, and tillage practice for the 17 PSUs were surveyed, and soil loss rate for each land use in the PSUs were estimated using an empirical model 18 Chinese Soil Loss Equation (CSLE). Though the information collected from the sample units can be aggregated to 19 estimate soil erosion conditions on a large scale, the problem of estimating soil erosion condition on a regional 20 scale has not been well addressed. The aim of this study is to introduce a new model-based regional soil erosion 21 assessment method combining sample survey and geostatistics. We compared fiveseven spatial interpolation 22 models based on Bivariate Penalized Spline over Triangulation (BPST) method to generate a regional soil erosion 23 assessment from the PSUs. Shaanxi province (3,116 PSUs) in China was used to conduct the comparison and 24 assessment as it is one of the areas with the most serious erosion problem. Ten-fold cross validation based on the 25 PSU data shownshowed Land use, rainfall erosivity, and soil erodibility at the resolution of 250×250 m pixels for 26 the entire domain were used as the auxiliary information. Shaanxi province (3,116 PSUs) in China was used to-27 conduct the comparison and assessment as it is one of the areas with the most serious erosion problem. The results showed three models with land use as the auxiliary information generated much lower mean squared errors (MSE) 28 29 than the other two models without land use. Tthe model assisted by the land use, rainfall erosivity factor (R), and

1 soil erodibility factor (K), slope steepness factor (S) and slope length factor (L) derived from 1:10000 topography 2 map is the best one, with the model efficiency coefficient (ME) being 0.75 and the which has-MSE being 55.8% 3 of that for less than half that of the model assisted by the land use alonesmoothing soil loss in the PSUs directly. 4 
[revised manuscript text omitted]
} \frac{\hat{Q}_{uj}}{\text{of}} Q_{uj}$  for every pixel *i*<del>obtain the</del>-9 interpolation over the domain. Then, Ffor anythe jth pixel in land use u, we estimate the soil loss  $A_{uj}$  by 10  $\hat{A}_{uj} = \hat{Q}_{uj} \cdot R_{uj},$ 11 (74)where  $\hat{T}_{uj}$  is the estimation of  $T_{uj}$  for the land use u and the pixel j. 12 Model III: Estimating A with K and land use as the auxiliary information. SThis model is similar towith 13 Model II, except that we -useing  $K_{ui}$  instead of  $R_{ui}$  in equations (3) and  $K_{uj}$  instead of  $R_{uj}$  in 14 15 equation (4). Model IV: Estimating A with L and land use as the auxiliary information. This model is sSimilar towith 16 Model II, except that we use using  $L_{ui}$  instead of  $R_{ui}$  in equations (3) and  $L_{uj}$  instead of  $R_{uj}$  in 17 18 equation (4). Model V: Estimating A with S and land use as the auxiliary information. This model is similar to Model II, 19 except that we use Similar with Model II, using  $S_{ui}$  instead of  $R_{ui}$  in equations (3) and  $S_{uj}$  instead of 20 21  $R_{uj}$  in equation (4). Model VI: Estimating A with R, K and land use as the auxiliary information. This model is similar to 22 Model II, except that we use Similar with Model II, using  $R_{ui}K_{ui}$  instead of  $R_{ui}$  in equations (3) and 23  $R_{uj}K_{uj}$  instead of  $R_{uj}$  in equation (4). 24

Model VII: Estimating A with R, K, L, S and land use as the auxiliary information. This model is similar to
Model II, except that we use Similar with Model II, using
$$R_{ui}K_{ui}L_{ui}S_{ui}$$
 instead of  $R_{ui}$  in equations (3)
and  $R_{uj}K_{uj}L_{uj}S_{uj}$  instead of  $R_{uj}$  in equation (4).
For land use u and sampling unit i, define
 $Q_{ui} = \frac{A_{ui}}{R_{ui} \cdot K_{ui}}$ , (8)

7 where  $A_{ui}$  is the soil erodibility value. For land use u, smoothing  $\mathcal{L}_{ui}$ 's over the domain, we obtain the 8 estimator  $\hat{\mathcal{Q}}_{uj}$  of  $\mathcal{Q}_{uj}$  for every pixel j. Then, for any jth pixel in land use u, we can estimate the soil loss  $A_{uj}$  by

**10 2.4.2 Bivariate penalized spline over triangulation method**

11 In spatial data analysis, there are mainly two approaches to make the prediction of a target variable. One 12 approach (e.g., kriging) treats the value of a target variable at each location as a random variable and uses the 13 covariance function between these random variables or a variogram to represent the correlation; another 14 approach (e.g., spline or wavelet smoothing) uses a deterministic smooth surface function to describe the 15 variations and connections among values at different locations. In this study, Bivariate Penalized Spline over 16 Triangulation (BPST), which belongs to the second approach, was used to explore the relationship between 17 location information in a two-dimensional (2-D) domain and the response variable. The BPST method we 18 consider in this work have has several advantages. First, it provides good approximations of smooth functions 19 over complicated domains. Second, the computational cost for spline evaluation and parameter estimation are 20 manageable. Third, the BPST doesn't require the data to be evenly distributed or on a -regular-spaced grid. 21 Since our data are a little sparse in some area, we employed the roughness penalties to regularize the spline fit; 22 see the energy functional defined in equation (12). When the sampling is sparse in certain area, the direct BPST 23 method may not be effective since the results may have high variability due to the small sample size. The 24 penalized BPST is more suitable for this type of data because it can help to regularize the fit.

1 To be more specific, let  $(x_i, y_i) \in \Omega$  be the latitude and longitude of unit i for i = 1, 2, ..., n. Suppose we 2 observe  $z_i$  at locations  $(x_i, y_i)$  and  $\{(x_i, y_i, z_i)\}_{i=1}^n$  satisfy 3  $z_i = f(x_i, y_i) + \epsilon_i, i = 1, 2, \dots, n,$ (105)where  $\varepsilon_i$ 's are random variables with mean zero, and f(.) is some smooth but unknown bivariate function. 4 5 To estimate f, we adopt the bivariate penalized splines overon triangulations to handle irregular domains. 6 In the following we discuss how to construct basis functions using bivariate splines on a triangulation of 7 the domain  $\Omega$ . Details of various facts about bivariate splines stated in this section can be found in Lai and 8 Schumaker (2007). See also Guillas and Lai (2010) and Lai and Wang (2013) for statistical applications of 9 bivariate splines on triangulations. 10 A triangulation of  $\Omega$  is a collection of triangles  $\Delta = \{\tau_1, \tau_2, ..., \tau_N\}$  whose union covers  $\Omega$ . In addition, if 11 a pair of triangles in  $\Delta$  intersects, then their intersection is either a common vertex or a common edge. For a 12 given triangulation  $\Delta$ , we can construct Bernstein basis polynomials of degree p separately on each triangle, and the collection of all such polynomials form a basis. In the following, let  $S_r^p(\Delta)$  be a spline 13 14 space of degree p and smoothness r over triangulation  $\Delta$ . Bivariate B-splines on the triangulation are 15 piecewise polynomials of degree p (polynomials on each triangle) that are smoothly connected across 16 common edges, in which the connection of polynomials on two adjacent triangles is considered smooth if directional derivatives up to the rth degree are continuous across the common edge. 17 18 To estimate f, we minimize the following penalized least square problem:  $\min_{f \in S_r^p(\Delta)} (z_i - f(x_i, y_i))^2 + \lambda PEN(f),$ 19 (116)20 Where  $\lambda$  is the roughness penalty parameter, and PEN(f) is the penalty given below:  $\text{PEN}(f) = \int_{\tau \in \Delta} \left( \frac{\partial^2 f(x,y)}{\partial x^2} \right)^2 + \left( \frac{\partial^2 f(x,y)}{\partial x \, \partial y} \right)^2 + \left( \frac{\partial^2 f(x,y)}{\partial y^2} \right)^2 dxdy,$ 21 (127)22 For Models I-VII defined in Section 2.4.1, we consider the above minimization to fit the model, and obtain

- 23 the smoothed surface using the measurements of data A (Models I and III) or Q (Models II and V) or T-
- 24 (Model IV) and their corresponding location information.

**25 2.5 Assessment methods**

- 26 MTo compare different models, ean squared prediction error (MSE) and Nash-Sutcliffe model efficiency
- 27 coefficient (ME) are used to assess the performance of models. wWe estimate the out-of-sample prediction
- 28 errors of each method using the tent-0-fold cross validation. We randomly split all the observations over the

entire domain (with the buffer zone) into ten roughly equal-sized parts. For each k-t= 1, 2, ...., 10, then we
leave out part kt, fit the model using to the other nine parts (combined) inside the boundary with the buffer zone,
and then obtain predictions for the left-out kth-tth part inside the boundary of Shaanxi Province. T-In the Model II
and Model II, MSEoverall is calculated as follows:-

- 5  $MSE_{overall} = \frac{\sum_{k=1}^{10} SSE_k}{n},$  (13)
- In Models III, IV and V, we consider land use as one covariate. Therefore, thehe overall mean squared
  prediction error (MSEoverall) is calculated by the average of the sum of the product of individual MSEu and the
  corresponding sample size. The overall MSEoverall was calculated as follows: wWe first calculated the MSE of
- 9 land each use u,  $u = 1, 2, \dots, \frac{811}{3}$ , similar as for Model I and Model II,

10
$$MSE_u = \frac{\sum_{tk=1}^{10} SSE_{tk}}{10n}$$
, (148)

11 Then, the overall MSE can be calculated using

12
$$MSE_{overall} = \frac{\sum_{u=1}^{118} MSE_u * C_u}{\sum_{u=1}^{811} C_u}.$$
 (159)

13 where  $C_u$  is the sample size for the land use u.

14 Model efficiency coefficient  $ME_u$  for the land use *u* is calculated as follows (Nash and Sutcliffe, 1970):

$$ME_{u} = 1 - \frac{\sum_{i}^{C_{u}} [A_{pre,u}(i) - A_{obs,u}(i)]^{2}}{\sum_{i}^{C_{u}} [A_{obs,u}(i) - \overline{A_{obs,u}}(i)]^{2}}$$
(10)

16 Apresimu (i) and Aobs,u (i) are the predictedsimulated and observed soil loss for the plot *i* for land use *u*. 17 MEoverall stands for the overall model efficiency by pooling all samples for different land uses together. The 18 ME compares the simulated and observed values relative to the line of perfect fit. The maximum possible 19 value of ME is 1, and the higher the value the better the model fit. An efficiency of ME < 0 indicates that 20 the mean of the observed soil loss is a better predictor of the data than the model. – 21 The soil loss rate is divided into sSix soil erosion intensity levels, were divided according to\_ the soil loss rate, 22 which were mild (less than 5 t ha-1y-1), slight (5-10 t ha-1y-1), moderate (10-20 t ha-1y-1), high (20-40 t ha-1y-1),

23 severe (40-80 t ha-1y-1), and extreme (no less than greater than 80 t ha-1y-1), 
[revised manuscript text omitted]

|               | Land use and sample size |                                     |                                   |               |                             |                             |                                     |                     | _                         |
|---------------|--------------------------|-------------------------------------|-----------------------------------|---------------|-----------------------------|-----------------------------|-------------------------------------|---------------------|---------------------------|
| Model         | Paddy             | Dry land & irrigated
land | Orchard
&
garden | Forest | Shrub
land | Grass
land | Constructio
n land | Bare
land | Over
all |
|               | 82                | 1048                                | 436                        | 1288          | 574                  | 684                  | 323                          | 32           | 4467               |
| LU            | -0.68                    | 0.34                                | 0.23                              | 0.20          | 0.60                        | 0.52                        | 0.06                                | 0.18                | 0.55                      |
| LU+R   | 0.05                     | 0.34                                | 0.23                              | 0.20          | 0.60                        | 0.53                        | 0.08                                | 0.24                | 0.55                      |
| LU+K   | -1.98                    | 0.41                                | 0.26                              | 0.24          | 0.67                        | 0.59                        | 0.08                                | 0.32                | 0.60                      |
| LU+L   | 0.15                     | 0.41                                | 0.31                              | 0.23          | 0.65                        | 0.62                        | 0.16                                | 0.22                | 0.59                      |
| LU+S   | -0.08                    | 0.46                                | 0.37                              | 0.23          | 0.65                        | 0.63                        | 0.26                                | 0.46                | 0.64                      |
| LU+R+K | -0.65                    | 0.41                                | 0.26                              | 0.24          | 0.68                        | 0.60                        | 0.10                                | 0.38                | 0.60                      |
| LU+R+K+L+S    | 0.82                     | 0.58                                | 0.40                              | 0.25          | 0.76                        | 0.75                        | 0.43                                | 0.97                | 0.75                      |

**2 Table 23. Soil loss rates (t ha-1y-1) for the farmland, forest, shrub land and grassland by Model VI in this study and in**

|                   | Land use                           | Mean                                              | Standard deviation            |
|-------------------|------------------------------------|---------------------------------------------------|-------------------------------|
| This study        | Dry land & irrigated land Farmland | 21 <del>19</del> . 77 <del>00</del> | 20 <del>17</del> .0694 |
|                   | Forest                             | 3 3. 51 50                          | 2 2. 77 78      |
|                   | Shrub land                         | 10 10.000                                  | 7 7. 51 51      |
|                   | Grassland                          | 7. 27 <del>20</del>                        | 5.2 0 <del>3</del>     |
| Guo et al. (2015) | Farmland (Conventional)            | 49.38                                             | 57.61                         |
|                   | Farmland (Ridge tillage)           | 19.27                                             | 13.35                         |
|                   | Farmland (Terracing)               | 0.12                                              | 0.28                          |
|                   | Forest                             | 0.10                                              | 0.12                          |
|                   | Shrub land                         | 8.06                                              | 7.47                          |
|                   | Grassland                          | 11.57                                             | 12.72                         |

3 Northwest region of China from Guo et al. (2015).

1 Table 34. Annual soil loss amount, mean rate and main sources by Model VI for ten prefecture cities in Shaanxi

|             |                               | Source (%)                    |                                       |                               |                   |                              |                    |
|-------------|-------------------------------|-------------------------------|---------------------------------------|-------------------------------|-------------------|------------------------------|--------------------|
| Prefecture  | Area (10 4 ha)     | Amount                        | Mean R rate                    | Dry land and                  |                   | Sheub                        | Cross              |
| city        |                               | $(10^6 \text{ t y}^{-1})$     | (t ha -1 y -1 ) | irrigated land                | Forest            |                              | land               |
|             |                               |                               |                                       | Farmland                      |                   | land                         | land               |
| Vilan       | 100 0100 4                    | ( 5 ( )                       | 6462                                  | 55 0 52 0                     | 11.2              | 7970                         | 19.6               |
| A1 an       | 100.9 100.4            | 0.3 0.3                | 0.4 <del>0.3</del> –           | 55.0 <del>52.9</del> - | <del>11.6  </del> |  1.0 <del>1.9</del> - | 20.6               |
| Ankong      | 224 1220 0                    | 27 1 26 6                     | 11 7 11 6                             | 167429                        | 9.4        | 2528                         | 38.5               |
| Ankang      | 254.1 <del>250.0</del> | 27.4 <del>26.6 -</del> | 11./ <del>11.6  </del>         | 46./ 42.8                     | <del>10.7  </del> | 2.5 <del>2.8</del> -  | 42.7               |
| Deeii       | 100 1170 5                    | 140120                        | 0 7 7 4                               | 26 4 20 2                     | 10.8              | 7275                         | 39.6        |
| Баојі       | 160.1 176.3            | 14.0 <del>13.2</del> - | 0.2 <del>/ .4</del> -          | 30.4 39.3              | <del>15.1</del>   |  1.3 <del>1.3</del> - | <del>37.9</del>    |
| Hanzhong    | 269 1266 7                    | 20.0.21.9                     | 7007                                  | 15 5 12 5                     | 11.4              | 3.2 <del>3.6</del>    | 36.5               |
| Halizholig  | 208.1 <del>200.7</del> | 20.9 <del>21.8 -</del> |  /.8 <del>8.2 -</del>          | 43.3 42.3              | 12.3              |                              | 40.2               |
| Shangluo    | 104 9102 0                    | 5.8  8.5 -             | 3.0 4.4                        | 38.3 68.0-             | 19.4              | 8.4 <del>5.9</del>    | 27.4               |
|             | 194.8 195.0            |                               |                                       |                               | <del>13.1</del>   |                              | <del>12.9</del>    |
| Tongohuan   | 29 929 6                      | 3037                          | 10 2 0 6                              | 40 1 37 0                     | 7.2               | 23.2                         | 28.2               |
| Toligenuali |  30.0 30.0             | 3.7 3.7                | 10.2 9.0                              | +0.1 57.5              | 7.8               | <del>23.6</del>              | 28.5               |
| Wainan      | 120 8120 5                    | 7564                          | 5750                                  | 50 6 54 4                     | 3.2               | 8805                         | 24.6               |
| vv eman     | 127.0127.3                    | 7.3 0.+                | 3.7 <del>3.0</del>             |                        | <del>3.9</del>    | 0.0 9.5                      | <del>26.7</del>    |
| Vienvena    | 102 8101 0                    | 5.6 <del>5.2</del>     | 5.5 <del>5.2</del>             | 46.3 44.4                     | 3.1               | 3.5 <del>8.9</del>    | 14.2               |
| Alanyang    | 102.0                         |                               |                                       |                               | 8.2               |                              | <del>35.3</del>    |
| Van'an      | 360 1364 0                    | 60 5 55 0                     | 16 / 15 3                             | 157515                        | 4.8               | 12.0                         | 37.0               |
| 1 an an     | 507.1 504.7            | 00.3 55.7                     | 10.4 19.5                             | 45.7 54.5                     | 3.1               | 12.1                         | <del>30.0   </del> |
| Vulin       | 122 7 727 7            | 56 5 50.0                     | 13/110                                | 56 3 51_4                     | 2.2               | 3637                         | 36.4               |
| 1 01111     |  722.1 721.1           |  30.3  <del>30.7</del> | 13.4 11.7 -                    | 50.5 51. <del>1</del>  | 2.6               | 3.0 <del>3.1</del>    | 40.4               |
| Overall     | 20/1 /2030 /                  | 207.3                         | 10 2 9 8                              | 10 2 10 8                     | 6.7               | 7171                         | 35.2               |
| Overall     | 2041.4 2030.4          | <del>198.7</del>              | 10.2 <del>9.8</del> -          | 49.2 <del>49.8 -</del> | <del>6.8</del>    | /.1_<del>/.1_</del>   | <del>35.0  </del>  |

**2 province.**

|                                            | Area
(10 4 ha) | Runoff
(10 9 m 3 y -1 ) | Sediment
discharge
(10 6 t y -1 ) | Soil loss
rate
(t ha -1 y -1 ) | Percent of
forest
(%) | Soil loss rate
for forest
(t ha -1 y -1 ) |
|--------------------------------------------|------------------------------|-------------------------------------------------------------|---------------------------------------------------------------|------------------------------------------------------------|-----------------------------|-----------------------------------------------------------------------|
| Jinghe ⁴ Jing
he ⁴ | 454.2                        | 1.837                                                       | 246.7                                                         | 54.3                                                       | 6.5                         | 19.0                                                                  |
| Luohe b Luo
he e  | 284.3                        | 0.906                                                       | 82.6                                                          | 29.1                                                       | 38.4                        | 1.3~2.1                                                               |

Table 4-5. Soil erosion rate for the forest and sediment discharge for two watersheds

2 d. Based on the observation at Zhangjiashan hydrological station from 1950 through 1989<del>.</del>

3 e. Based on the observation of at Zhuanghe hydrological station from 1959 through 1989.

1

---

## Author Response (AR3)

Dear Professor Axel Bronstert,

Dear reviewers,

Thank you for handling and reviewing our manuscript (hess-2016-394). We are glad to hear that you and the reviewers are positive to our manuscript and we appreciate all your comments, which help the manuscript a lot. We dealt carefully with all the comments and took suggestions under consideration one by one. All changes have been made using 'track changes' in MS Word.

Reviewer 1# paid special attention to the comparison of the method proposed in our study with the raster layer multiplication method used in the recent European water erosion assessment (Panagos et al., 2015b). There were two comments relating to this issue (Comment 1: Authors did not convince both reviewers that it is better to use their proposal instead of applying an USLE with the available resolution of input data; Comment 2: General comment about the methodology: Interpolating the soil erosion modelled point estimates is less appropriate than the application of model using input layers. Authors do not make clear statements of this and readers may be confused in making interpolation of erosion points even if they have the input layers. This point should be better addressed. ) When the resolution of input data for the entire region is good enough to derive all the erosion factors, we agree with the reviewer that the raster multiplication approach would be better than the method proposed in this study. However, in many practical situation it may not be possible to obtain all the erosion factors at high resolution.   For example, our analysis shows that the topography factor is very important to the soil erosion estimation and the requirements on the DEM resolution to derive the reliable topography factors are high, which can be not met in many countries at this stage. The proposed approach uses all the factor information that are available in the entire region (land use, rainfall, soils), and uses spatial interpolation to impute other factor information which are only available at the sampled PSU, which provides a viable alternative which is of practical importance. Section 4.2 addresses these two comments: "*Raster multiplication is a popular model-based approach due to its lower cost, simpler procedures and easier explanation of resulting map.*

*If the resolution of input data for the entire region is enough to derive all the erosion*

*factors, raster multiplication approach is the best choice. However, there are several*

*concerns about raster multiplication approach for two reasons: (1) The information for the*

*support practices factor (P) in the USLE was not easy to collect given the common image*

*resolution and was not included in some assessments (Lu et al., 2001; Rao et al., 2015), in*

*which the resulting maps don't reflect the condition of soil loss but the risk of soil loss.*

*Without the information of P factor, it is also impossible to assess the benefit from the soil*

*and water conservation practices (Liu et al., 2013). (2) The accuracy of soil erosion*

*estimation for each cell is of concern if the resolution of database used to derive the*

*erosion factors is limited. For example, The LS-factor in the new assessment of soil loss by*

*water erosion in Europe (Panagos et al., 2015b) was calculated using the 25-m DEM,*

*which may result in some errors for the entire region due to the limited resolution of DEM*

*data for each cell (Wang et al., 2016). In this study, the information we can get at this stage*

*for the entire region is land use, rainfall erosivity (R) and soil erodibility (K). The other*

*factors were not available or without enough resolution. It is not difficult to conduct raster*

*layer multiplication technically, however, we think the multiplication of R and K factors*

*(assuming L=1, S=1, B=1, E=1, T=1) reflects the potential of soil erosion, which is*

*different from the soil erosion estimated in this study. Therefore, we did not compare our*

*method with raster layer multiplication method. Our recommended approach uses all the*

*factor information that are available in the entire region (land use, rainfall, soils), and uses*

*spatial interpolation to impute other factor information which are only available at the*

*sampled PSU (slope degree, slope length, practice and management, aggregated as Q) to*

*the entire region. The rationale behind this approach is to exploit the spatial dependence*

*among these factors to come up with better regional estimates. Since the reality in many*

*countries is that we cannot have all factors measured in all areas in the foreseeable future,*

*or the resolution of data for deriving the factors is limited, we believe our approach*

*provides a viable alternative which is of practical importance. "*

The other comments raised by Reviewer 1# and responses were listed thereunder:

(1) The erosion rates in forest lands are very extreme and not justified. This can be connected to the erroneous outputs from NDVI calculations (I am very concerned about the NDVI. This is also confirmed by the difference with the study of Guo.

**Response**: The soil loss rate for the forest in this study was 3.51 t ha$^{-1}$ y$^{-1}$ with a standard deviation of 2.77 t ha$^{-1}$ y$^{-1}$ , which is much higher than 0.10 t ha$^{-1}$ y$^{-1}$ reported in Guo et al. (2015, Table 3) based on plot observations. However, observations on the runoff and sediment discharge for two watersheds have demonstrated that the soil loss rate for the forest in the study area has large variability ranging from 1.3 to 19.0 t ha$^{-1}$ y$^{-1}$ (Fig. 1, Table 5, Wang and Fan, 2002). Our estimation is within the range. We discussed the possible reasons for the soil loss rate for the forest in detail (Section 4.2). We were also concerned about the quality of the NDVI data. We do find some bias when comparing the NDVI data with the observations of vegetation cover in the sample units, however, we had corrected the bias according to the observations before we used the NDVI data to calculate the B factor. To sum up, we don't think the quality of NDVI is the reason for the erosion rate in forest lands because the bias has been already corrected before it was used in this study.

(2) The whole issue with LS factor and how authors present it. Previous studies in European LS-factor and the study of Borrelli in soil erosion in Italy demonstrated that the finer the resolution, the best are the LS-factor estimates and soil erosion model outputs. I do not agree with the what authors state in P12L13-14. The same confusion is created in P17L20-23. In the past studies in Europe it was shown that the finer the resolution, the best representation of the topography and then the more accurate the estimate of LS-factor and erosion rates.

**Response**: we agree with the reviewer that the finer the resolution, the better are the LS-factor estimates and soil erosion model outputs. P12L13-14 reports "*Previous research suggested topography factors should be derived from high resolution topography information (such as 1:10000 or larger scale topography contour map, Thomas et al., 2015). Topography factors based on smaller scale of topography map (such as 1:50000 or 30-m DEM) in the mountainous and hilly area have large uncertainties (Wang et al., 2016).*" , which is consistent with the reviewer's opinion. To avoid the confusion, we revised the sentence into *"Previous research has suggested topography factors should be*

*derived from high resolution topography information (such as 1:10000 or topography*

*contour maps with finer resolutions, Thomas et al., 2015). Topography factors based on*

*topography maps with coarser resolutions (such as 1:50000 or 30-m DEM) in the*

*mountainous and hilly areas have large uncertainties (Wang et al., 2016).* ”

P17L20-23 reports: “ *For example, The LS-factor in the new assessment of soil loss by*

*water erosion in Europe (Panagos et al., 2015b) was calculated using the 25-m DEM,*

*which may result in some errors for the entire region due to the limited resolution of DEM*

*data for each cell (Wang et al., 2016).* ”, to be more specific, we revise it to be “*For*

*example, The LS-factor in the new assessment of soil loss by water erosion in Europe*

*(Panagos et al., 2015b) was calculated using the 25-m DEM, which may result in some*

*errors for the mountainous and hilly areas due to the limited resolution of DEM data for*

*each cell (Wang et al., 2016).*”

(3) Table 1: quite confusion. What are the numbers in the row below the land uses?
Moreover model numbers are missing in this table

**Response**: The numbers in the row below the land uses are the sample size for each land
use, which are noted in the revision. The model numbers were also added to the table.

(4) The low importance given in the model approach for R-factor is preoccupying.

**Response**: The R-factor in our study region has relatively small spatial variability, which may be the reason for the low importance of R factor as the covariate. The comparison of four models with single erosion factor as the covariate (Model II, III, IV and V) showed S

factor is the best covariate, with $MSE_{overall}$ for Model V being 80.1% of that for Model I, whereas R is the worst, with $MSE_{overall}$ for Model II being 99.3% of that for Model I

(5) P2L15-15: The 2 sentences on water & wind erosion are repetitive

**Response**: The repetitive sentence was deleted.

(6) P2L20. I don't agree with this sentence. It maybe that land use is improved and the soil erosion risk decreased.

**Response:** It is true that the risk may decrease in the future. The sentence was revised into "*Assessments on the risks of soil erosion under different scenarios of climate change and land use are also very important (Kirkby et al., 2008).*"

(7) P3L3. Experts have always expertise

**Response:** We agree with the reviewer on it.

(8) P3L29: "seamless" is not appropriate.

**Response:** It was revised into "*The second category is based on the multiplication of erosion factor raster layers.*"

(9) P5L3-5: The sentence is confusing

**Response:** It was revised into "*While robust and reliable for large domains which contain enough sample sites, such method has large uncertainties when it was used for the small domain. The method to obtain domain level statistics used in the fourth census of soil erosion in China was different from that used by NRI. A simple spatial model was used to smooth the proportion of soil erosion directly in China, which is an attempt to interpolate sample survey units information using geostatistics.*"

(10) P6L25L: for first time you present the factors R, K, L, S….B, E: and the reader does not know what those factors are. In the next page you describe them…but there should be a logical sequence.

**Response:** Good suggestion. We added abbreviations where these factors appear for the first time. See blue words in the following sentences:

"*The erosion factors rainfall erosivity (R) and soil erodibility (K) are also available for the entire domain. The slope length (L) and slope degree (S) factors can be derived from 30-m and 90-m Digital Elevation Model (DEM) data from shuttle radar topography mapping mission (SRTM). The other factors including the biological (B), engineering (E) and tillage (T) practice factors are either impossible or very difficult to obtain for the entire region at this stage.*"

(11) Eq (1): What is uk? Describe below..

**Response:** u denotes the land use type, k denotes the $k^{th}$ plot with the land use u in a sample unit. They are defined in the revision.

(12) P7L27: soil families are not a good term

**Response:** We think soil family is a proper term here. "USDA soil taxonomy (ST) developed by United States Department of Agriculture and the National Cooperative Soil Survey provides an elaborate classification of soil types according to several parameters (most commonly their properties) and in several levels: Order, Suborder, Great Group, Subgroup,        Family,        and        Series.        (cited        from https://en.wikipedia.org/wiki/USDA_soil_taxonomy) "

(13) Equation 2: What is q?

**Response:** q is the number of plots with the land use u in the unit i. It was added in the revision.

(14)What are the measurement units of factors in equation 2?

**Response:** They were added.

(15) L10P17: What is f(,)

**Response:** It is $f(x_i,y_i)$. It has been revised.

(16)Tabl3e 5: What not provide the total soil loss and discuss on the comparison of soil loss with sediment loss?

**Response:** The sediment discharges from the watershed outlets can be measured. Sediment delivery ratio, the ratio of sediment discharge from the watershed outlet to the total soil loss, usually equals or is less than one. It was assumed to be one in this study for the steep terrain in the watersheds. Then the total soil loss equals the sediment discharge. Table 5 is to show the observations of soil loss rate for the forest in the study area. We think the comparison of soil loss with sediment discharge is not within the scope of the discussion. However, an annotation was added in the Table 5: "*Sediment delivery ratio, the ratio of sediment discharge from the watershed outlet to the total soil loss, was assumed to be one; Soil loss rate was defined as the soil loss per unit area.*"

The comments raised by Reviewer 2# and responses were list below:

(1) This paper addresses an interesting issue and is generally well presented. However, there is potential for improving it by including some further statistics for assessing model performance in addition to the mean square error of prediction (MSE) and the model efficiency, such as the bias and the correlation. It is surprising that there is such a marked deterioration in the MSE when coarser resolution topography data are used and these additional statistics might provide some insight into this.

**Response:** In addition to the tables of MSE and model efficiency, a scatterplot (Figure 6) to show the deviation of the simulation from the observation for four main land uses was presented, which demonstrated that there was no obvious bias for these four main land uses. Since we already have five tables and ten figures, and adding new tables of bias and correlation does not provide additional insights, we decided not to add new tables and figures to show the bias and the correlation. Instead, we added the following notes under the MSE table: "*Since Figure 6 showed no obvious systematic bias for four main land uses, we didn't list the bias separately in this table.*"

(2) SSE in equation 8 should be defined as the sum of squared prediction errors.
**Response:** Added!

(3) A few minor corrections are needed to the English e.g. 'have been' instead of 'were' on page 2 line 25; 'needs' instead of 'need' on page 2 line 28; add 'the' before 'factorial scoring approach' on page 3 line 4 and 'a' before 'Plot-based approach' on page 3 line 5.
**Response:** Revised, as well as some other grammar mistakes.

We strongly believe that the revision process has led to a significant improvement of the manuscript. If you have any further questions, please don't hesitate to contact us.

Sincerely,

Baoyuan Liu

[revised manuscript text omitted]